# Convex Relaxation for Solving Large-Margin Classifiers in Hyperbolic Space

**Sheng Yang**  *shengyang@g.harvard.edu*
*John A. Paulson School of Engineering and Applied Sciences*
*Harvard University*

**Peihan Liu**  *peihanliu@fas.harvard.edu*
*John A. Paulson School of Engineering and Applied Sciences*
*Harvard University*

**Cengiz Pehlevan**  *cpehlevan@seas.harvard.edu*
*John A. Paulson School of Engineering and Applied Sciences*
*Center for Brain Science*
*Kempner Institute for the Study of Natural and Artificial Intelligence*
*Harvard University*

**Reviewed on OpenReview:** *https://openreview.net/forum?id=eIPwJgadfZ*

## Abstract

Hyperbolic spaces have increasingly been recognized for their outstanding performance in handling data with inherent hierarchical structures compared to their Euclidean counterparts. However, learning in hyperbolic spaces poses significant challenges. In particular, extending support vector machines to hyperbolic spaces is in general a constrained non-convex optimization problem. Previous and popular attempts to solve hyperbolic SVMs, primarily using projected gradient descent, are generally sensitive to hyperparameters and initializations, often leading to suboptimal solutions. In this work, by first rewriting the problem into a polynomial optimization, we apply semidefinite relaxation and sparse moment-sum-of-squares relaxation to effectively approximate the optima. From extensive empirical experiments, these methods are shown to achieve better classification accuracies than the projected gradient descent approach in most of the synthetic and real two-dimensional hyperbolic embedding dataset under the one-vs-rest multiclass-classification scheme.

## 1 Introduction

The $d$-dimensional hyperbolic space $\mathbb{H}^d$ is the unique simply-connected Riemannian manifold with a constant negative sectional curvature -1. Its exponential volume growth with respect to radius motivates representation learning of hierarchical data using the hyperbolic space. Representations embedded in the hyperbolic spaces have demonstrated significant improvements over their Euclidean counterparts across a variety of datasets, including images (Khrulkov et al., 2020), natural languages (Nickel & Kiela, 2017), and complex tabular data such as single-cell sequencing (Klimovskaia et al., 2020).

On the other hand, learning and optimization on hyperbolic spaces are typically more involved than that on Euclidean spaces. Problems that are convex in Euclidean spaces become constrained non-convex problems in hyperbolic spaces. The hyperbolic Support Vector Machine (HSVM), as explored in recent studies Cho et al. (2019); Chien et al. (2021), exemplifies such challenges by presenting as a non-convex constrained programming problem that has been solved predominantly based on projected gradient descent. Attempts have been made to alleviate its non-convex nature through reparametrization (Mishne et al., 2023) or developing a hyperbolic perceptron algorithm that converges to a separator with finetuning using adversarial samples to approximate the large-margin solution (Weber et al., 2020). To our best knowledge, these attempts are

grounded in the gradient descent dynamics, which is highly sensitive to initialization and hyperparameters and cannot certify optimality.

As efficiently solving for the large-margin solution on hyperbolic spaces to optimality provides performance gain in downstream data analysis, we explore two convex relaxations to the original HSVM problem and examine their empirical tightness through their optimality gaps. Our contributions can be summarized as follows: in Section 3, we briefly introduce the necessary concepts for large-margin learning in hyperbolic space, transform the original HSVM formulation into a quadratically constrained quadratic programming (QCQP) problem, and later apply the standard semidefinite relaxation (SDP) (Shor, 1987) to this QCQP. Empirically, SDP does not yield tight enough solutions, which motivates us to apply the moment-sum-of-squares relaxation (Moment) (Nie, 2023). The problem with the Moment approach is its limited scalability as it empirically requires a long runtime and large computation memory. However, there is a special star-shaped sparsity structure in the HSVM problem. By exploiting such a sparcity pattern, we successfully reduce the number of decision variables in the original moment-sum-of-squares relaxation and propose an equivalent but sparse moment-sum-of-squares relaxation. In Section 4, we test the performance of our methods in both simulated and real datasets, We observe small optimality gaps for various tasks (in the order of $10^{-2}$ to $10^{-1}$) by using the sparse moment-sum-of-squares relaxation and obtain better max-margin separators in terms of test accuracy in a one-vs-rest 5-fold train-test scheme than projected gradient descent (PGD). SDP relaxation, on the other hand, is not tight, but still yields better solutions than PGD, particularly in the one-vs-one training framework. The section then ends with the limitations of our proposed method and a brief practical guidance of when to use our methods. Lastly, we conclude and point out some future directions in Section 5. Additionally, we propose without testing a robust version of HSVM in Appendix F. The code to our implmentations is https://github.com/yangshengaa/hsvm-relax.

## 2 Related Works

Support Vector Machine (SVM) is a classical statistical learning algorithm operating on Euclidean features Cortes & Vapnik (1995). This convex quadratic optimization problem aims to find a linear separator that classifies samples of different labels and has the largest margin to data samples. The problem can be efficiently solved through coordinate descent or Lagrangian dual with sequential minimal optimization (SMO) Platt (1998) in the kernelized regime. Mature open source implementations exist such as `LIBLINEAR` Fan et al. (2008) for the former and `LIBSVM` Chang & Lin (2011) for the latter.

Less is known when moving to statistical learning on non-Euclidean spaces, such as hyperbolic spaces. The popular practice is to directly apply neural networks in both obtaining the hyperbolic embeddings and perform inferences, such as classification, on these embeddings Ganea et al. (2018); Klimovskaia et al. (2020); Nickel & Kiela (2017); Chami et al. (2020; 2019); Lensink et al. (2022); Skliar & Weiler (2023); Shimizu et al. (2020); Peng et al. (2021). Recently, increasing attention has been paid to transferring standard Euclidean statistical learning techniques, such as SVMs, to hyperbolic embeddings for both benchmarking neural net performances and developing better understanding of inherent data structures Mishne et al. (2023); Weber et al. (2020); Cho et al. (2019); Chien et al. (2021). Learning a large-margin solution on hyperbolic space, however, involves a non-convex constrained optimization problem. Cho et al. (2019) propose and solve the hyperbolic support vector machine problem using projected gradient descent; Weber et al. (2020) add adversarial training to gradient descent for better generalizability; Chien et al. (2021) propose applying Euclidean SVM to features projected to the tangent space of a heuristically-searched point to bypass PGD; Mishne et al. (2023) reparametrize parameters and features back to Euclidean space to make the problem nonconvex and perform normal gradient descent. All these attempts are, however, gradient-descent-based algorithms, which are sensitive to initialization, hyperparameters, and class imbalances, and can provably converge to a local minimum without a global optimality guarantee.

Another relevant line of research focuses on providing efficient convex relaxations for various optimization problems, such as using semidefinite relaxation (Shor, 1987) for QCQP and moment-sum-of-squares (Blekherman et al., 2012) for polynomial optimization problems. The flagship applications of SDP includes efficiently solving the max-cut problem on graphs Goemans & Williamson (1995) and more recently in machine learning tasks such as rotation synchronization in computer vision (Eriksson et al., 2018), robotics (Rosen et al.,

2020), and medical imaging (Wang & Singer, 2013). Some results on the tightness of SDP have been analyzed on a per-problem basis (Bandeira et al., 2017; Brynte et al., 2022; Zhang, 2020). On the other hand, moment-sum-of-squares relaxation, originated from algebraic geometry (Blekherman et al., 2012; Lasserre, 2001), has been studied extensively from a theoretical perspective and has been applied for certifying positivity of functions in a bounded domain (Henrion & Lasserre, 2005). Synthesizing the work done in the control and algebraic geometry literature and geometric machine learning works is under-explored.

## 3 Convex Relaxation Techniques for Hyperbolic SVMs

In this section, we first introduce fundamentals on hyperbolic spaces and the original formulation of the hyperbolic Support Vector Machine (HSVM) due to Cho et al. (2019). Next, we present two relaxations techniques, the semidefinite relaxation and the moment-sum-of-squares relaxation, that can be solved efficiently with convergence guarantees. Our discussions center on the Lorentz manifold as the choice of hyperbolic space, since it has been shown in Mishne et al. (2023) that the Lorentz formulation offers greater numerical advantages in optimization.

### 3.1 Preliminaries

**Hyperbolic Space (Lorentz Manifold):** define *Minkowski product* of two vectors $\mathbf{x}, \mathbf{y} \in \mathbb{R}^{d+1}$ as $\boldsymbol{x} * \boldsymbol{y} = x_0 y_0 - \sum_{i=1}^{d} x_i y_i$. A $d$-dimensional hyperbolic space (Lorentz formulation) is a submanifold embedded in $\mathbb{R}^{d+1}$ defined by,

$$\mathbb{H}^d := \{\boldsymbol{x} = (x_0, \boldsymbol{x}) \in \mathbb{R}^{d+1} | \; \boldsymbol{x} * \boldsymbol{x} = 1, x_0 > 0\}. \tag{1}$$

**Tangent Space:** a tangent space to a manifold at a given point $\boldsymbol{x} \in \mathbb{H}^d$ is the local linear subspace approximation to the manifold, denoted $T_{\boldsymbol{x}} \mathbb{H}^d$. In this case the tangent space is a Euclidean vector space of dimension $d$ written as

$$T_{\boldsymbol{x}} \mathbb{H}^d = \{\boldsymbol{w} \in \mathbb{R}^{d+1} | \; \boldsymbol{w} * \boldsymbol{x} = 0\}. \tag{2}$$

**Exponential & Logarithmic Map:** the exponential map $\exp_{\boldsymbol{x}}(.) : T_{\boldsymbol{x}} \mathbb{H}^d \to \mathbb{H}^d$ is a transformation that sends vectors in the tangent space to the manifold. The logarithmic map $\log_{\boldsymbol{x}}(.) : \mathbb{H}^d \to T_{\boldsymbol{x}} \mathbb{H}^d$ is the inverse operation. Formally, given $\boldsymbol{x} \in \mathbb{H}^d, \boldsymbol{v} \in T_{\boldsymbol{x}} \mathbb{H}^d$, we have

$$\exp_{\boldsymbol{x}}(\boldsymbol{v}) = \cosh(\|\boldsymbol{v}\|_{\mathbb{H}^d})\boldsymbol{x} + \sinh(\|\boldsymbol{v}\|_{\mathbb{H}^d})\frac{\boldsymbol{v}}{\|\boldsymbol{v}\|_{\mathbb{H}^d}} \; , \;\; \|\boldsymbol{v}\|_{\mathbb{H}^d} = \sqrt{-\boldsymbol{v} * \boldsymbol{v}}. \tag{3}$$

Exponential and logarithmic maps serve as bridges between Euclidean and hyperbolic spaces, enabling the transfer of notion, such as distances and probability distributions, between these spaces. One way is to consider Euclidean features as residing within the tangent space of the hyperbolic manifold's origin. From this standpoint, distributions on hyperbolic space can be obtained through $\exp_{\mathbf{0}}$.

**Hyperbolic Decision Boundary:** straight lines in the hyperbolic space are intersections between $d$-dimensional hyperplanes passing through the origin and the manifold $\mathbb{H}^d$. Suppose $\boldsymbol{w} \in \mathbb{R}^{d+1}$ is the normal direction of the plane, then the plane and hyperbolic manifold intersect if and only if $\boldsymbol{w} * \boldsymbol{w} < 0$. From this viewpoint, each straight line in the hyperbolic space can be parameterized by $\boldsymbol{w}$ and can be considered a linear separator for hyperbolic embeddings. Hence, we can define a decision function $h_{\boldsymbol{w}}(.)$, by the Minkowski product of the feature

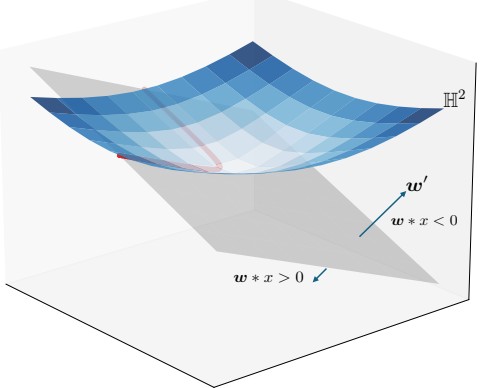

Figure 1: Straight line (red) on Lorentz manifold $\mathbb{H}^2$ as the intersection between a hyperplane and the manifold, presented similarly in Cho et al. (2019).

with the decision plane, as the following,

$$h_{\boldsymbol{w}}(\boldsymbol{x}) = \begin{cases} 1, & \boldsymbol{w} * \boldsymbol{x} = -(\boldsymbol{w}')^T \boldsymbol{x} > 0, \\ -1, & \text{otherwise}, \end{cases} \tag{4}$$

where $\boldsymbol{w}' = [-w_0, w_1, ..., w_d]$. A visualization is presented in Figure 1.

**Stereographic Projection:** we visualize $\mathbb{H}^2$ by projecting Lorentz features isometrically to the Poincaré space $\mathbb{B}^d$. Denote Lorentz features as $\boldsymbol{x} = [x_0, x_1, ..., x_d]$, then its projection is given by $\tilde{\boldsymbol{x}} = [\frac{x_1}{1+x_0}, ..., \frac{x_d}{1+x_0}] \in \mathbb{B}^d \subset \mathbb{R}^d$. Decision boundaries on the Lorentz manifold are mapped to arcs in the Poincaré space. The proof is deferred to Appendix A.2.

## 3.2 Original Formulation of the HSVM

Cho et al. (2019) proposed the hyperbolic support vector machine which finds a max-margin separator where margin is defined as the hyperbolic point to line distance. We demonstrate our results in a binary classification setting. Extension to multi-class classification is straightforward using Platt-scaling (Platt et al., 1999) in the one-vs-rest scheme or majority voting in one-vs-one setting.

Suppose we are given $\{(\boldsymbol{x}_i, y_i) : \boldsymbol{x}_i \in \mathbb{H}^d, y_i \in \{1, -1\}\}_{i=1}^n$. The hard-margin HSVM is formulated as,

$$(\textbf{HARD}) \quad \min_{\boldsymbol{w} \in \mathbb{R}^{d+1}, \boldsymbol{w}^T G \boldsymbol{w} > 0} \frac{1}{2} \boldsymbol{w}^T G \boldsymbol{w} \quad \text{s.t.} \; -y_i(\boldsymbol{x}_i^T G \boldsymbol{w}) \geqslant 1, \forall i \in [n], \tag{5}$$

whereas the soft-margin version allows misclassification using

$$(\textbf{SOFT}) \quad \min_{\boldsymbol{w} \in \mathbb{R}^{d+1}, \boldsymbol{w}^T G \boldsymbol{w} > 0} \frac{1}{2} \boldsymbol{w}^T G \boldsymbol{w} + C \sum_{i=1}^n l(-(y_i(G\boldsymbol{x}_i))^T \boldsymbol{w}), \tag{6}$$

where $G \in \mathbb{R}^{(d+1) \times (d+1)}$ is a diagonal matrix with diagonal elements $\texttt{diag}(G) = [-1, 1, 1, ..., 1]$ (i.e. all ones but the first being -1), to represent the Minkowski product in a Euclidean matrix-vector product manner and is the source of indefiniteness of the problem. In the soft-margin case, the hyperparameter $C \geqslant 0$ controls the strength of penalizing misclassification. This penalty scales with hyperbolic distances, defined by $l(z) = \max(0, \text{arcsinh}(1) - \text{arcsinh}(z))$.

As $C$ approaches infinity, we recover the hard-margin formulation from the soft-margin one. In the rest of the paper we focus on analyzing relaxations to the soft-margin formulation in Equation (6) as these relaxations can be applied to both hyperbolic-linearly separable or unseparable data.

To solve the problem efficiently, we have two observations that lead to two adjustments in our approach. Firstly, although the constraint involving $\boldsymbol{w}$ is initially posited as a strict inequality, practical considerations allow for a relaxation. Specifically, when equality is achieved, $\boldsymbol{w}^T G \boldsymbol{w} = 0$, the separator is not on the manifold and assigns the same label to all data samples. However, with sufficient samples for each class in the training set and an appropriate regularization constant $C$, the solver is unlikely to default to such a trivial solution. Therefore, we may substitute the strict inequality with a non-strict one during implementation. Secondly, the penalization function, $l$, is not a polynomial. Although projected gradient descent is able to tackle non-polynomial terms in the loss function, solvers typically only accommodate constraints and objectives expressed as polynomials. We thus take a Taylor expansion of the arcsinh term to the first order so that every term in the formulation is a polynomial. This also helps with constructing our semidefinite and moment-sum-of-squares relaxations later on, which is presented in Appendix A.1 in detail. The new formulation of the soft-margin HSVM outlined in Equation (6) is then given by,

$$\min_{\boldsymbol{w} \in \mathbb{R}^{d+1}, \xi \in \mathbb{R}^n} \frac{1}{2} \boldsymbol{w}^T G \boldsymbol{w} + C \sum_{i=1}^n \xi_i \quad ,$$

$$\text{s.t.} \; \xi_i \geqslant 0, \forall i \in [n] \tag{7}$$

$$(y_i(G\boldsymbol{x}_i))^T \boldsymbol{w} \leqslant \sqrt{2}\xi_i - 1, \forall i \in [n]$$

$$\boldsymbol{w}^T G \boldsymbol{w} \geqslant 0$$

where $\xi_i$ for $i \in [n]$ are the slack variables. More specifically, given a sample $(\boldsymbol{x}_i, y_i)$, if $\xi_i = 0$, the sample has been classified correctly with a large margin; if $\xi_i \in (0, \frac{1}{\sqrt{2}}]$, the sample falls into the right region but with a small hyperbolic margin; and if $\xi_i > \frac{1}{\sqrt{2}}$, the sample sits in the wrong side of the separator. We defer a detailed derivation of Equation (7) to Appendix A.1 and its extension to curvatures other than $-1$ in Appendix A.3.

### 3.3 Semidefinite Formulation

Note that Equation (7) is a non-convex quadratically-constrained quadratic programming (QCQP) problem, we can apply a semidefinite relaxation (SDP) (Shor, 1987). The SDP formulation is given by

$$
\textbf{(SDP)} \min_{\substack{\boldsymbol{W} \in \mathbb{R}^{(d+1)\times(d+1)} \\ \boldsymbol{w} \in \mathbb{R}^{d+1} \\ \xi \in \mathbb{R}^n}} \frac{1}{2}\mathbf{Tr}(G, \boldsymbol{W}) + C \sum_{i=1}^{n} \xi_i \quad ,
$$

$$
\begin{aligned}
\text{s.t.} \quad & \xi_i \geqslant 0, \forall i \in [n] \\
& (y_i(G\boldsymbol{x}_i))^T \boldsymbol{w} \leqslant \sqrt{2}\xi_i - 1, \forall i \in [n] \\
& \mathbf{Tr}(G, \boldsymbol{W}) \geqslant 0 \\
& \begin{bmatrix} 1 & \boldsymbol{w}^T \\ \boldsymbol{w} & \boldsymbol{W} \end{bmatrix} \succeq 0
\end{aligned} \tag{8}
$$

where decision variables are highlighted in bold and that the last constraint stipulates the concatenated matrix being positive semidefinite, which is equivalent to $\boldsymbol{W} - \boldsymbol{w}\boldsymbol{w}^T \succeq 0$ by Schur's complement lemma. In this SDP relaxation, all constraints and the objective become linear in $(\boldsymbol{W}, \boldsymbol{w}, \xi)$, which could be easily solved. Note that if additionally we mandate $\boldsymbol{W}$ to be rank 1, then this formulation would be equivalent to Equation (7) or otherwise a relaxation. Moreover, it is important to note that this SDP does not directly yield decision boundaries. Instead, we need to extract $\boldsymbol{w}^*$ from the solutions $(\boldsymbol{W}, \boldsymbol{w}, \xi)$ obtained from Equation (8). A detailed discussion of the extraction methods is deferred to Appendix B.1.

### 3.4 Moment-Sum-of-Squares Relaxation

The SDP relaxation in Equation (8) may not be tight, particularly when the resulting $\boldsymbol{W}$ has a rank much larger than 1. Indeed, we often find $\boldsymbol{W}$ to be full-rank empirically. In such cases, moment-sum-of-squares relaxation may be beneficial. Specifically, it can certifiably find the global optima, provided that the solution exhibits a special structure, known as the flat-extension property (Curto & Fialkow, 2005; Henrion & Lasserre, 2005).

We begin by introducing some necessary notions, with a more comprehensive introduction available in Appendix C. We define the relaxation order as $\kappa \geqslant 1$ and our decision variables as $\boldsymbol{q} = (\boldsymbol{w}, \xi) \in \mathbb{R}^{n+d+1}$. Our objective, $p(\boldsymbol{q})$, is a polynomial of degree $2\kappa$ with input $\boldsymbol{q}$, where its coefficient is defined such that $p(\boldsymbol{q}) = \frac{1}{2}\boldsymbol{w}^T G \boldsymbol{w} + C \sum_{i=1}^{n} \xi_i$, thus matching the original objective. Hence, the polynomial $p(\cdot)$ has $s(m, 2\kappa) := \binom{m+2\kappa}{2\kappa}$ number of coefficients, where $m = n + d + 1$ is the dimension of decision variables. Additionally, we define $z \in \mathbb{R}^{s(m,2\kappa)}$ as the *Truncated Multi-Sequence* (TMS) of degree $2\kappa$, and we denote a linear functional $f$ associated with this sequence as

$$
f_{\boldsymbol{z}}(p) = \langle f_{\boldsymbol{z}}, p \rangle = \langle z, \text{vec}(p) \rangle, \tag{9}
$$

which is the inner product between the coefficients of polynomial $p$ and the vector or real numbers $\boldsymbol{z}$. The vector of monomials up to degree $\kappa$ generated by $\boldsymbol{q}$ is denoted as $[\boldsymbol{q}]_\kappa$. With all these notions established, we can then define the **moment matrix** of $\kappa$-th degree, $M_\kappa[\boldsymbol{z}]$, and **localizing matrix** of $\kappa$-th degree for polynomial $g$, $L_{\kappa,g}[\boldsymbol{z}]$, as the followings,

$$
M_\kappa[\boldsymbol{z}] = \langle f_{\boldsymbol{z}}, [\boldsymbol{q}]_\kappa [\boldsymbol{q}]_\kappa^T \rangle, \tag{10}
$$

$$
L_{\kappa,g}[\boldsymbol{z}] = \langle f_{\boldsymbol{z}}, g(\boldsymbol{q}) \cdot [\boldsymbol{q}]_s [\boldsymbol{q}]_s^T \rangle, \tag{11}
$$

where $s$ is the max degree such that $2s + \deg(g) \leqslant 2\kappa$, $[\boldsymbol{q}]_\kappa [\boldsymbol{q}]_\kappa^T$ is a matrix of polynomials with size $s(m, \kappa)$ by $s(m, \kappa)$, and all the inner products are applied element-wise above. For example, if $n = 1$ and $d = 2$ (i.e. 1 data sample from a 2-dimensional hyperbolic space), the degree-2 monomials generated by $\boldsymbol{q} = (w_0, w_1, w_2, \xi_1)$ are

$$[\boldsymbol{q}]_2^T = [1, w_0, w_1, w_2, \xi_1, w_0^2, w_1^2, w_2^2, \xi_1^2, w_0 w_1, w_0 w_2, w_0 \xi_1, w_1 w_2, w_1 \xi_1, w_2 \xi_1]. \tag{12}$$

With all these definitions established, we can present the moment-sum-of-squares relaxation (Nie, 2023) to the HSVM problem, outlined in Equation (7), as

$$\begin{aligned} (\textbf{Moment}) \quad & \min_{\boldsymbol{z} \in \mathbb{R}^{s(m, 2\kappa)}} \langle \text{vec}(p), \boldsymbol{z} \rangle \quad . \\ & s.t. \ M_\kappa[\boldsymbol{z}] \succeq 0 \\ & \quad L_{\kappa, \xi_i}[\boldsymbol{z}] \succeq 0, \ \forall i \in [n] \\ & \quad L_{\kappa, -(y_i(Gx_i))^T \boldsymbol{w} + \sqrt{2} \xi_i - 1}[\boldsymbol{z}] \succeq 0, \ \forall i \in [n] \\ & \quad L_{\kappa, \boldsymbol{w}^T G \boldsymbol{w}}[\boldsymbol{z}] \succeq 0 \end{aligned} \tag{13}$$

Note that $g(\boldsymbol{q}) \geqslant 0$, as previously defined, serves as constraints in the original formulation. Additionally, when forming the moment matrix, the degree of generated monomials is $s = \kappa - 1$, since all constraints in Equation (7) has maximum degree 1. Consequently, Equation (13) is a convex programming and can be implemented as a standard SDP problem using mainstream solvers. We further emphasize that by progressively increasing the relaxation order $\kappa$, we can find increasingly better solutions theoretically, as suggested by Lasserre (2018).

However, moment-sum-of-squares relaxation does not scale with the data size due to the combinatorial factors in the dimension of truncated multi-sequence $\boldsymbol{z}$, leading to prohibitively slow runtimes and excessive memory consumption. To address this issue, we exploit the sparsity pattern inherent in this problem: many generated monomial terms do not appear in the objective or constraints. For instance, there is no cross-terms among the slack variables, such as $\xi_i \xi_j$ for $i \neq j \in [n]$. Specifically, in this problem, we observe a **star-shaped sparsity** structure, as ilustrated in Figure 2. We observe that, by defining sparsity groups as $\boldsymbol{q}^{(i)} = (\boldsymbol{w}, \xi_i)$, two nice structural properties can be found: first, the objective function involves all the sparsity groups, $\{\boldsymbol{q}^{(i)}\}_{i=1}^n$ and can be decomposed as sum of polynomials $p^{(i)}$, which involves only $\boldsymbol{q}^{(i)}$ and thus has a smaller number of coefficients $s(m', 2\kappa)$, where $m' = d + 2$. Second, each constraint is exclusively associated with a single group $\boldsymbol{q}^{(i)}$ for a specific $i$. For

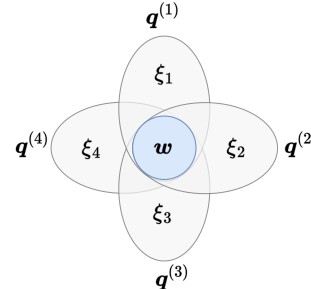

Figure 2: Star-shaped Sparsity pattern in Equation (13) with $n = 4$

the remaining constraint, $\boldsymbol{w}^T G \boldsymbol{w}$, we could assign it to group $i = 1$ without loss of generality. Hence, by leveraging this sparsity property, we can reformulate the moment-sum-of-squares relaxation equivalently into its sparse version,

$$\begin{aligned} (\textbf{Sparse-Moment}) \quad & \min_{\boldsymbol{z}^{(i)} \in \mathbb{R}^{s(m', 2\kappa)}, \forall i \in [n]} \sum_{i=1}^n \langle \text{vec}(p^{(i)}), \boldsymbol{z}^{(i)} \rangle \quad , \\ & s.t. \ M_\kappa[\boldsymbol{z}^{(i)}] \succeq 0, \forall i \in [n] \\ & \quad L_{\kappa, \xi_i}[\boldsymbol{z}^{(i)}] \succeq 0, \ \forall i \in [n] \\ & \quad L_{\kappa, -(y_i(Gx_i))^T \boldsymbol{w} + \sqrt{2} \xi_i - 1}[\boldsymbol{z}^{(i)}] \succeq 0, \ \forall i \in [n] \\ & \quad L_{\kappa, \boldsymbol{w}^T G \boldsymbol{w}}[\boldsymbol{z}^{(1)}] \succeq 0 \\ & \quad (M_\kappa[\boldsymbol{z}^{(i)}])_{kl} = (M_\kappa[\boldsymbol{z}^{(1)}])_{kl}, \forall i \geqslant 2, (k, l) \in B \end{aligned} \tag{14}$$

where $B$ is an index set of the moment matrix to entries generated by $\boldsymbol{w}$ along, ensuring that each moment matrix with overlapping regions share the same values as required. We refer the last constraint as the sparse-binding constraint. The nice thing about Equation (14) is that it is equivalent to Equation (13) and has a fewer number of decision variables.

Unfortunately, our solution empirically does not satisfy the flat-extension property and we cannot not certify global optimality. Nonetheless, in practice, it achieves significant performance improvements in selected datasets over both projected gradient descent and the SDP-relaxed formulation. Similarly, this formulation does not directly yield decision boundaries and we defer discussions on the extraction methods to Appendix B.2.

## 4 Experiments

We validate the performances of **semidefinite relaxation (SDP)** and **sparse moment-sum-of-squares relaxations (Moment)** by comparing various metrics with that of **projected gradient descent (PGD)** on a combination of synthetic and real datasets. The PGD implementation follows from adapting the MATLAB code in Cho et al. (2019), with learning rate 0.001 and 2000 epochs for synthetic and 4000 epochs for real dataset and warm-started with a Euclidean SVM solution.

**Datasets.** For synthetic datasets, we construct Gaussian and tree embedding datasets following Cho et al. (2019); Mishne et al. (2023); Weber et al. (2020). Regarding real datasets, our experiments include two machine learning benchmark datasets, CIFAR-10 Krizhevsky et al. (2009) and Fashion-MNIST Xiao et al. (2017) with their hyperbolic embeddings obtained through standard hyperbolic embedding procedure (Chien et al., 2021; Khrulkov et al., 2020; Klimovskaia et al., 2020) to assess image classification performance. Additionally, we incorporate three graph embedding datasets, such as football, karate, and polbooks obtained from Chien et al. (2021), to evaluate the effectiveness of our methods on graph-structured data. We also explore cell embedding datasets, including Paul Myeloid Progenitors developmental dataset (Paul et al., 2015), Olsson Single-Cell RNA sequencing dataset (Olsson et al., 2016), Krumsiek Simulated Myeloid Progenitors dataset(Krumsiek et al., 2011), and Moignard blood cell developmental trace dataset from single-cell gene expression (Moignard et al., 2015), where the inherent geometry structures well fit into our methods.

We emphasize that all features are on the Lorentz manifold, but visualized in Poincaré manifold through stereographic projection if the dimension is 2.

**Evaluation Metrics.** The primary metrics for assessing model performance are average training and testing loss, accuracy, and weighted F1 score under a stratified 5-fold train-test split scheme. Furthermore, to assess the tightness of the relaxations, we examine the **relative suboptimality gap**, defined as

$$\eta = \frac{|\hat{f} - p^*|}{1 + |p^*| + |\hat{f}|}, \tag{15}$$

where $f^*$ is the unknown optimal objective value, $p^*$ is the objective value of the relaxed formulation, and $\hat{f}$ is the objective associated to the max-margin solution recovered from the relaxed model. Clearly $p^* \leqslant f^* \leqslant \hat{f}$, so if $\eta \approx 0$, we can certify the exactness of the relaxed model.

**Implementations Details.** We use `MOSEK` (ApS, 2022) in `Python` as our optimization solver without any intermediate parser, since directly interacting with solvers save substantial runtime in parsing the problem. `MOSEK` uses interior point method to update parameters inside the feasible region without projections. All experiments are run and timed on a machine with 8 Intel Broadwell/Ice Lake CPUs and 40GB of memory. Results over multiple random seeds have been gathered and reported.

We first present the results on synthetic Gaussian and tree embedding datasets in Section 4.1, followed by results on various real datasets in Section 4.2.

### 4.1 Synthetic Dataset

**Synthetic Gaussian.** To generate a Gaussian dataset on $\mathbb{H}^d$, we first generate Euclidean features in $\mathbb{R}^d$ and lift to hyperbolic space through exponential map at the origin, $\exp_0$, as outlined in Equation (3). We adjust the number of classes $K \in \{2, 3, 5\}$ and the variance of the isotropic Gaussian $s \in \{0.4, 0.6, 0.8, 1.0\}$.

Three Gaussian embeddings in $d = 2$ are selected and visualized in Figure 3 and performances with $C = 10$ for the three dataset are summarized in Table 1.

In general, we observe a small gain in average test accuracy and weighted F1 score from SDP and Moment relative to PGD. Notably, we observe that Moment often shows more consistent improvements compared to SDP, across most of the configurations. In addition, Moment gives smaller optimality gaps $\eta$ than SDP. This matches our expectation that Moment is tighter than the SDP.

Although in some case, for example when $K = 5$, Moment achieves significantly smaller losses compared to both PGD and SDP, it is generally not the case. We emphasize that these losses are not direct measurements of the max-margin hyperbolic separators' generalizability; rather, they are combinations of margin maximization and penalization for misclassification that scales with $C$. Hence, the observation that the performance in test accuracy and weighted F1 score is better, even though the loss computed using extracted solutions from SDP and Moment is sometimes higher than that from PGD, might be due to the complicated loss landscape. More specifically, the observed increases in loss can be attributed to the intricacies of the landscape rather than the effectiveness of the optimization methods. Based on the accuracy and F1 score results, empirically SDP and Moment methods identify solutions that generalize better than those obtained by running gradient descent alone. We provide a more detailed analysis on the effect of hyperparameters in Appendix E.2 and runtime in Table 5. Decision boundary for Gaussian 1 is visualized in Figure 5 as an example.

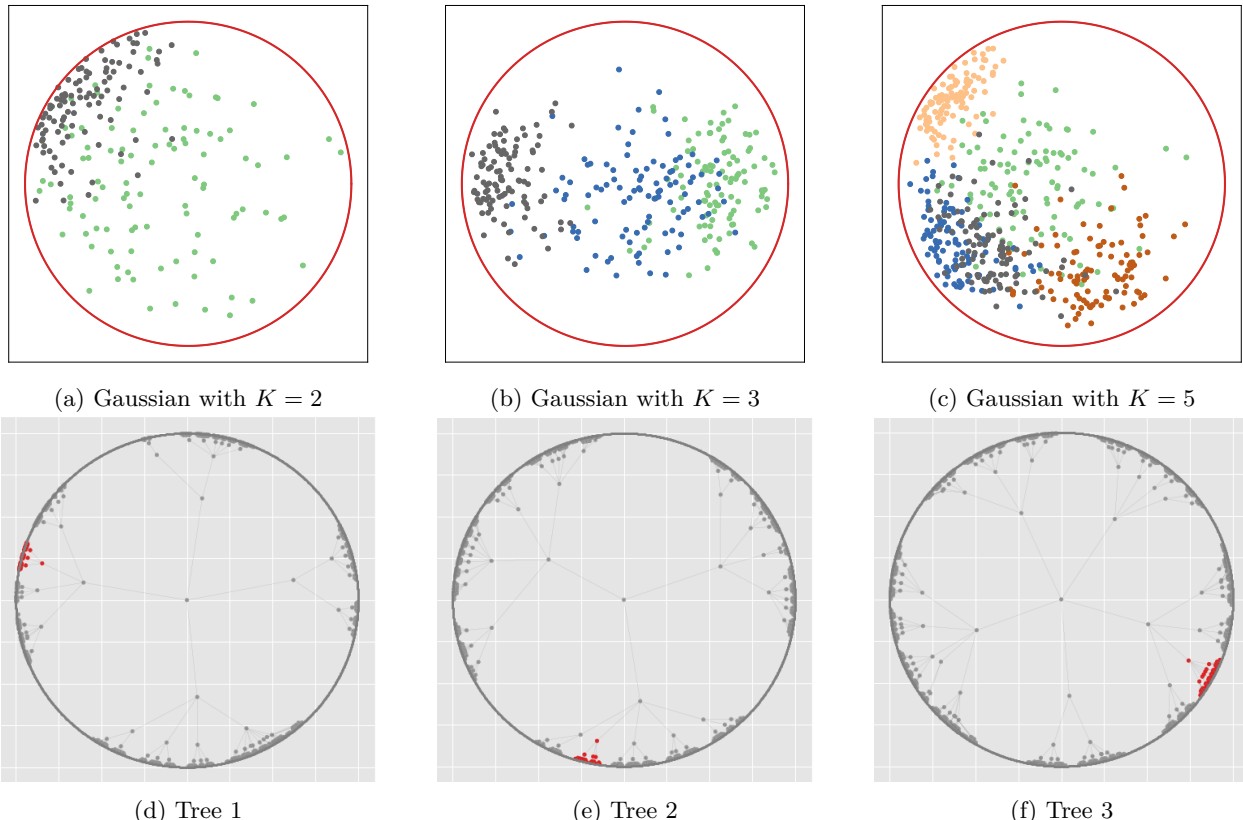

(a) Gaussian with $K = 2$      (b) Gaussian with $K = 3$      (c) Gaussian with $K = 5$

(d) Tree 1      (e) Tree 2      (f) Tree 3

Figure 3: Three Synthetic Gaussian (top row) and Three Tree Embeddings (bottom row). All features are in $\mathbb{H}^2$ but visualized through stereographic projection on $\mathbb{B}^2$. Different colors represent different classes. For tree dataset, the graph connections are also visualized but not used in training. The selected tree embeddings come directly from Mishne et al. (2023).

**Synthetic Tree Embedding.** As hyperbolic spaces are good for embedding trees, we generate random tree graphs and embed them to $\mathbb{H}^2$ following Mishne et al. (2023). Specifically, we label nodes as positive if they are children of a specified node and negative otherwise. Our models are then evaluated for subtree classification, aiming to identify a boundary that includes all the children nodes within the same subtree.

Such task has various practical applications. For example, if the tree represents a set of tokens, the decision boundary can highlight semantic regions in the hyperbolic space that correspond to the subtrees of the data graph. We emphasize that a common feature in such subtree classification task is data imbalance, which usually lead to poor generalizability. Hence, we aim to use this task to assess our methods' performances under this challenging setting. Three embeddings are selected and visualized in Figure 3 and performance is summarized in Table 1. The runtime of the selected trees can be found in Table 5. Decision boundary of tree 2 is visualized in Figure 6.

Similar to the results of synthetic Gaussian datsets, we observe better performance from SDP and Moment compared to PGD, and due to data imbalance that gradient-based methods typically struggle with, we have a larger gain in weighted F1 score in this case. In addition, we observe large optimality gaps for SDP but very tight gap for Moment, certifying the optimality of Moment even when class-imbalance is severe.

Table 1: Performance on synthetic Gaussian and tree dataset for $C = 10.0$: 5-fold test accuracy and weighted F1 score plus and minus 1 standard deviation, and the average relative optimality gap $\eta$ for SDP and Moment.

| data | test acc | | | test f1 (micro) | | | $\eta$ | |
|---|---|---|---|---|---|---|---|---|
| | **PGD** | **SDP** | **Moment** | **PGD** | **SDP** | **Moment** | **SDP** | **Moment** |
| gaussian 1 | $84.50\% \pm 7.31\%$ | $\mathbf{85.50\%} \pm \mathbf{8.28\%}$ | $\mathbf{85.50\%} \pm \mathbf{8.28\%}$ | $0.84 \pm 0.07$ | $\mathbf{0.85} \pm \mathbf{0.08}$ | $\mathbf{0.85} \pm \mathbf{0.08}$ | 0.0847 | **0.0834** |
| gaussian 2 | $85.33\% \pm 4.88\%$ | $84.00\% \pm 5.12\%$ | $\mathbf{86.33\%} \pm \mathbf{4.76\%}$ | $0.86 \pm 0.05$ | $0.84 \pm 0.06$ | $\mathbf{0.87} \pm \mathbf{0.05}$ | 0.2046 | **0.0931** |
| gaussian 3 | $75.8\% \pm 3.31\%$ | $72.80\% \pm 3.37\%$ | $\mathbf{77.40\%} \pm \mathbf{2.65\%}$ | $0.75 \pm 0.03$ | $0.71 \pm 0.04$ | $\mathbf{0.77} \pm \mathbf{0.03}$ | 0.2204 | **0.0926** |
| tree 1 | $96.11\% \pm 2.95\%$ | $\mathbf{100.0\%} \pm \mathbf{0.00\%}$ | $\mathbf{100.0\%} \pm \mathbf{0.00\%}$ | $0.94 \pm 0.04$ | $\mathbf{1.00} \pm \mathbf{0.00}$ | $\mathbf{1.00} \pm \mathbf{0.00}$ | 0.9984 | **0.0640** |
| tree 2 | $96.25\% \pm 0.00\%$ | $99.71\% \pm 0.23\%$ | $\mathbf{99.91\%} \pm \mathbf{0.05\%}$ | $0.94 \pm 0.00$ | $1.00 \pm 0.00$ | $\mathbf{1.00} \pm \mathbf{0.00}$ | 0.9985 | **0.0205** |
| tree 3 | $99.86\% \pm 0.16\%$ | $99.86\% \pm 0.16\%$ | $\mathbf{99.93\%} \pm \mathbf{0.13\%}$ | $0.99 \pm 0.00$ | $0.99 \pm 0.00$ | $\mathbf{0.99} \pm \mathbf{0.00}$ | 0.3321 | **0.0728** |

## 4.2 Real Dataset

Real datasets consist of embedding of various sizes and number of classes in $\mathbb{H}^2$, visualized in Figure 4. We first report performances of three models using one-vs-rest training scheme, described in Appendix D, in Tables 6 to 8 for $C \in \{0.1, 1.0, 10\}$ respectively, and report aggregated performances, by selecting the one with the highest average test weighted F1 score, in Table 2. In general, we observe that Moment achieves the best test accuracy and weighted F1 score, particularly in biological datasets with clear hyperbolic structures, and have smaller optimality gaps compared to SDP relaxation, for nearly all selected data. However, it is important to note that the optimality gaps of these two methods remain distant from zero, suggesting that these relaxations are not tight enough for these datasets. Nevertheless, both relaxed models significantly outperform projected gradient descent (PGD) by a wide margin. Furthermore, our observations reveal that in the one-vs-rest training scheme, PGD shows considerable sensitivity to the choice of the regularization parameter $C$ from Tables 6 to 8, whereas SDP and Moment are less affected, demonstrating better stability and consistency across different $C$'s.

One critical drawback of semidefinite and sparse moment-sum-of-squares relaxation is that they do not scale efficiently with an increase in data samples, resulting in excessive consumption of time and memory, for example, CIFAR10 and Fashion-MNIST using a one-vs-rest training scheme. The workaround is one-vs-one training scheme, where we train for $\mathcal{O}(K^2)$ number of classifiers among data from each pair of classes and make final prediction decision using majority voting. We summarize the performance in Table 3 by aggregating results for different $C$ in Tables 9 to 11 as in the one-vs-rest case. We observe that in one-vs-one training, the improvement in general from the relaxation is not as significant as it in the one-vs-rest scheme, and SDP relaxation now gives the best performance in average test accuracy and test F1, albeit with large optimality gaps. Note that in the one-vs-one scheme, PGD is more consistent across different $C$'s, potentially because each subproblem-binary classifying one class against another-contains less data compared to one-vs-rest, making it easier to identify solutions.

A more detailed analysis on the effect of regularization $C$ and runtime comparisons are provided in Appendix E.3 and Table 12.

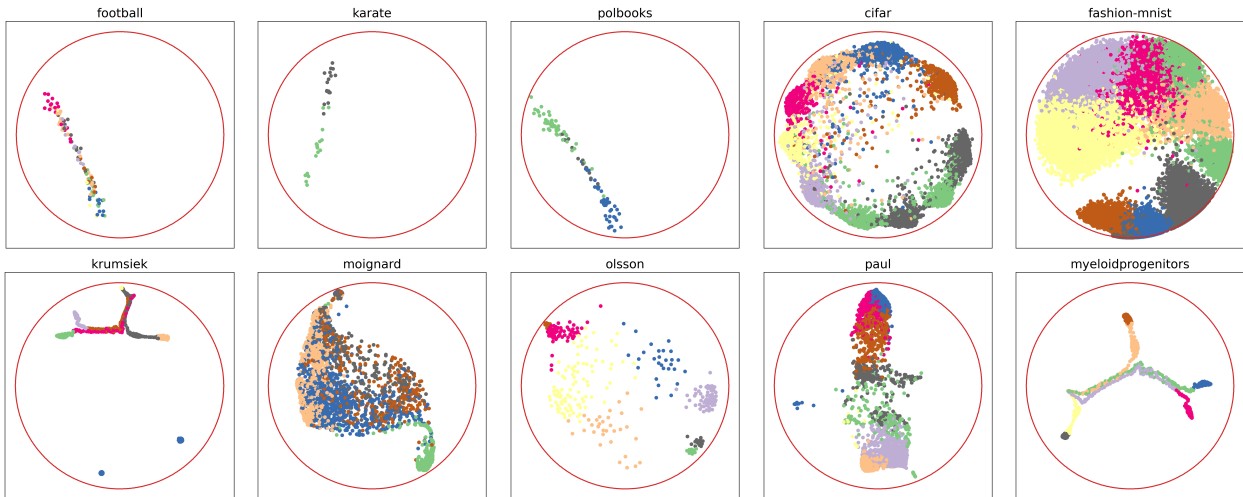

Figure 4: Real datasets embedded on $\mathbb{H}^2$ visualized in $\mathbb{B}^2$. Different colors represent different classes. The first three (football, karate, polbooks) are graph embeddings; the latter two (cifar10, fashion mnist) on the top row are standard ML benchmarks; the last 5 dataset are single-cell sequencing data embedded on $\mathbb{H}^2$ for cell type discovery and miscellaneous biomedical usages.

Table 2: Real dataset 5-fold test accuracy, F1, and optimality gap with one-vs-rest training. Best metrics based on weighted F1 is reported here aggregating from Tables 6 to 8.

| data | test acc | | | test f1 (micro) | | | $\eta$ | |
| | PGD | SDP | Moment | PGD | SDP | Moment | SDP | Moment |
|---|---|---|---|---|---|---|---|---|
| football | **40.87% ± 4.43%** | 32.17% ± 4.43% | 37.39% ± 4.43% | **0.29 ± 0.03** | 0.23 ± 0.04 | 0.26 ± 0.03 | 0.3430 | **0.0999** |
| karate | 50.00% ± 6.39% | 50.00% ± 6.39% | 50.00% ± 6.39% | 0.66 ± 0.06 | 0.66 ± 0.06 | 0.66 ± 0.06 | 0.9155 | **0.0818** |
| polbooks | 84.76% ± 1.90% | 84.76% ± 1.90% | 84.76% ± 1.90% | 0.79 ± 0.02 | **0.80 ± 0.03** | **0.80 ± 0.03** | 0.1711 | **0.0991** |
| krumsiek | 81.78% ± 2.66% | 82.56% ± 2.01% | **86.47% ± 0.64%** | 0.79 ± 0.03 | 0.80 ± 0.03 | **0.84 ± 0.00** | 0.7519 | **0.0921** |
| moignard | 63.37% ± 0.70% | 63.68% ± 1.75% | **63.78% ± 1.57%** | **0.62 ± 0.01** | 0.60 ± 0.02 | 0.60 ± 0.02 | **0.0325** | 0.0396 |
| olsson | 74.27% ± 5.15% | 79.63% ± 3.54% | **81.20% ± 3.68%** | 0.69 ± 0.07 | 0.77 ± 0.04 | **0.79 ± 0.04** | 0.4118 | **0.0976** |
| paul | 54.85% ± 1.26% | 53.72% ± 2.42% | **64.71% ± 2.36%** | 0.48 ± 0.02 | 0.47 ± 0.03 | **0.61 ± 0.02** | 0.4477 | **0.0861** |
| myeloidprogenitors | 69.34% ± 3.81% | 70.12% ± 3.28% | **76.84% ± 2.04%** | 0.66 ± 0.05 | 0.67 ± 0.04 | **0.75 ± 0.02** | 0.6503 | **0.1074** |

## 4.3 Limitations

Although our proposed SDP and sparse moment relaxation methods achieve generally better classification performance compared to the PGD baseline in the one-vs-rest setting – and slightly less so in the one-vs-one setting – the computational cost is significant due to their longer runtimes, as previously mentioned. One naturally suspects that given equivalent computational resources (including time and memory), PGD might achieve performance comparable to the sparse moment relaxation methods. This scenario becomes especially

Table 3: Real dataset 5-fold test accuracy, F1, and optimality gap with one-vs-one training. Best metrics based on weighted F1 is reported here aggregating from Tables 9 to 11.

| data | test acc | | | test f1 (micro) | | | $\eta$ | |
| | PGD | SDP | Moment | PGD | SDP | Moment | SDP | Moment |
|---|---|---|---|---|---|---|---|---|
| football | 40.00% ± 5.07% | **42.61% ± 5.07%** | 41.74% ± 7.06% | 0.32 ± 0.06 | **0.35 ± 0.06** | 0.33 ± 0.07 | 0.6699 | **0.2805** |
| karate | 50.00% ± 6.39% | 50.00% ± 6.39% | 50.00% ± 6.39% | 0.34 ± 0.07 | 0.34 ± 0.07 | 0.34 ± 0.07 | 0.9986 | **0.0921** |
| polbooks | 83.81% ± 3.81% | **86.67% ± 1.90%** | 83.81% ± 2.33% | 0.80 ± 0.04 | **0.84 ± 0.03** | 0.81 ± 0.03 | 0.3383 | **0.1051** |
| krumsiek | 89.76% ± 0.80% | **90.46% ± 1.18%** | 90.38% ± 1.49% | **0.90 ± 0.01** | **0.90 ± 0.01** | 0.90 ± 0.02 | 0.5843 | **0.3855** |
| moignard | **63.50% ± 1.35%** | 62.53% ± 1.10% | 62.66% ± 1.17% | **0.62 ± 0.01** | 0.61 ± 0.01 | 0.61 ± 0.01 | **0.0312** | 0.0401 |
| olsson | 93.40% ± 2.75% | 94.03% ± 2.12% | **94.36% ± 0.78%** | 0.93 ± 0.03 | 0.94 ± 0.02 | **0.94 ± 0.01** | 0.9266 | **0.2534** |
| paul | 66.98% ± 2.89% | **68.85% ± 2.26%** | 68.52% ± 2.39% | 0.64 ± 0.03 | **0.66 ± 0.02** | 0.66 ± 0.03 | 0.7863 | **0.2130** |
| myeloidprogenitors | 79.81% ± 2.00% | 80.28% ± 2.50% | **80.60% ± 2.58%** | 0.80 ± 0.02 | 0.80 ± 0.02 | **0.81 ± 0.02** | 0.8911 | **0.1960** |
| cifar | 98.38% ± 0.14% | **98.42% ± 0.17%** | **98.42% ± 0.17%** | 0.98 ± 0.00 | 0.98 ± 0.00 | 0.98 ± 0.00 | 0.0825 | **0.0550** |
| fashion-mnist | 94.42% ± 1.10% | **95.28% ± 0.16%** | 95.23% ± 0.15% | 0.94 ± 0.01 | **0.95 ± 0.00** | **0.95 ± 0.00** | 0.3492 | **0.0054** |

plausible considering the large hyperparameter space available for PGD: a more aggressive grid search on learning rates, total number of epochs, regularization parameters, and initializations could potentially yield results that rival or even surpass those of the moment-based method.

To explore this further, we provide an additional comparison between our methods and PGD under approximately equal computational time constraints using the one-vs-rest scheme. Specifically, according to Table 5, the average runtime per training fold for the Moment method is approximately two orders of magnitude larger than that for PGD. Therefore, we randomly generated 100 different initializations for PGD, with each initialization vector drawn uniformly from the unit sphere subject to the condition $\boldsymbol{w}^T G \boldsymbol{w} > 0$, and report the best performance among these random initializations in Table 4 below.

Table 4: Synthetic and real dataset accuracy, F1, and optimality gap with one-vs-rest training. PGD (RANDOM INIT) results are the best among 100 random initializations.

| data | test acc | | | | test f1 (micro) | | | | $\eta$ | |
|---|---|---|---|---|---|---|---|---|---|---|
| | PGD | PGD (random init) | SDP | Moment | PGD | PGD (random init) | SDP | Moment | SDP | Moment |
| tree 1 | $96.11\% \pm 2.95\%$ | $93.63\% \pm 0.23\%$ | $\mathbf{100.0\%} \pm \mathbf{0.00\%}$ | $\mathbf{100.0\%} \pm \mathbf{0.00\%}$ | $0.94 \pm 0.04$ | $0.91 \pm 0.00$ | $\mathbf{1.00} \pm \mathbf{0.00}$ | $\mathbf{1.00} \pm \mathbf{0.00}$ | $0.9984$ | $\mathbf{0.0640}$ |
| tree 2 | $96.25\% \pm 0.00\%$ | $96.25\% \pm 0.01\%$ | $99.71\% \pm 0.23\%$ | $\mathbf{99.91\%} \pm \mathbf{0.05\%}$ | $0.94 \pm 0.00$ | $0.94 \pm 0.00$ | $\mathbf{1.00} \pm \mathbf{0.00}$ | $\mathbf{1.00} \pm \mathbf{0.00}$ | $0.9985$ | $\mathbf{0.0205}$ |
| tree 3 | $99.86\% \pm 0.16\%$ | $96.07\% \pm 3.21\%$ | $99.86\% \pm 0.16\%$ | $\mathbf{99.93\%} \pm \mathbf{0.13\%}$ | $0.99 \pm 0.00$ | $0.94 \pm 0.05$ | $0.99 \pm 0.00$ | $0.99 \pm 0.00$ | $0.3321$ | $\mathbf{0.0728}$ |
| football | $\mathbf{40.87\%} \pm \mathbf{4.43\%}$ | $39.13\% \pm 6.74\%$ | $32.17\% \pm 4.43\%$ | $37.39\% \pm 4.43\%$ | $\mathbf{0.29} \pm \mathbf{0.03}$ | $0.27 \pm 0.06$ | $0.23 \pm 0.04$ | $0.26 \pm 0.03$ | $0.3430$ | $\mathbf{0.0999}$ |
| karate | $50.00\% \pm 6.39\%$ | $50.00\% \pm 6.39\%$ | $50.00\% \pm 6.39\%$ | $50.00\% \pm 6.39\%$ | $0.66 \pm 0.06$ | $0.34 \pm 0.07$ | $0.66 \pm 0.06$ | $0.66 \pm 0.06$ | $0.9155$ | $\mathbf{0.0818}$ |
| polbooks | $84.76\% \pm 1.90\%$ | $83.81\% \pm 2.33\%$ | $84.76\% \pm 1.90\%$ | $84.76\% \pm 1.90\%$ | $0.79 \pm 0.02$ | $0.79 \pm 0.03$ | $\mathbf{0.80} \pm \mathbf{0.03}$ | $\mathbf{0.80} \pm \mathbf{0.03}$ | $0.1711$ | $\mathbf{0.0991}$ |
| krumsiek | $81.78\% \pm 2.66\%$ | $81.63\% \pm 1.71\%$ | $82.56\% \pm 2.01\%$ | $\mathbf{86.47\%} \pm \mathbf{0.64\%}$ | $0.79 \pm 0.03$ | $0.78 \pm 0.02$ | $0.80 \pm 0.03$ | $\mathbf{0.84} \pm \mathbf{0.00}$ | $0.7519$ | $\mathbf{0.0921}$ |
| moignard | $63.37\% \pm 0.70\%$ | $60.45\% \pm 1.02\%$ | $63.68\% \pm 1.75\%$ | $63.78\% \pm 1.57\%$ | $\mathbf{0.62} \pm \mathbf{0.01}$ | $0.58 \pm 0.01$ | $0.60 \pm 0.02$ | $0.60 \pm 0.02$ | $\mathbf{0.0325}$ | $0.0396$ |
| olsson | $74.27\% \pm 5.15\%$ | $76.17\% \pm 5.38\%$ | $79.63\% \pm 3.54\%$ | $\mathbf{81.20\%} \pm \mathbf{3.68\%}$ | $0.69 \pm 0.07$ | $0.73 \pm 0.06$ | $0.77 \pm 0.04$ | $\mathbf{0.79} \pm \mathbf{0.04}$ | $0.4118$ | $\mathbf{0.0976}$ |
| paul | $54.85\% \pm 1.26\%$ | $54.12\% \pm 1.60\%$ | $53.72\% \pm 2.42\%$ | $\mathbf{64.71\%} \pm \mathbf{2.36\%}$ | $0.48 \pm 0.02$ | $0.47 \pm 0.02$ | $0.47 \pm 0.03$ | $\mathbf{0.61} \pm \mathbf{0.02}$ | $0.4477$ | $\mathbf{0.0861}$ |
| myeloidprogenitors | $69.34\% \pm 3.81\%$ | $69.02\% \pm 3.36\%$ | $70.12\% \pm 3.28\%$ | $\mathbf{76.84\%} \pm \mathbf{2.04\%}$ | $0.66 \pm 0.05$ | $0.65 \pm 0.04$ | $0.67 \pm 0.04$ | $\mathbf{0.75} \pm \mathbf{0.02}$ | $0.6503$ | $\mathbf{0.1074}$ |

It's worth noting that for PGD, we follow Cho et al. (2019), where the initialization $w$ is determined based on the solution $w'$ of a soft-margin SVM in the ambient Euclidean space of the Lorentz model. This choice typically ensures better initialization, contributing to the stability of optimization, especially when numerous local optima may exist. Consequently, we observed that the best PGD performance among 100 different random initializations do not significantly improve its performance in this regime. Nevertheless, we do not exclude the possibility that more strategic hyperparameter tuning under similar computational constraints might yield improved PGD performance

Based on these insights, we offer practical recommendations for method selection tailored to dataset characteristics and scale. For small-scale, low-dimensional hyperbolic datasets, particularly in 2D, where a hyperparameter-free solution is desired, the SDP and sparse moment relaxation methods clearly excel. Additionally, we advocate exhibiting significant class overlap. For instance, arguably Gaussian 1 and 3 have greater overlap compared to Gaussian 2 (Figure 3), leading to a notable performance drop in the PGD method. We suspect that more mixing likely leads PGD to poor local minima, although this conjecture warrants further quantitative investigation. For larger dataset such as MNIST or comparable modern machine learning benchmarks, in which generalization to unseen data takes priority over strict global optimality, we recommend PGD due to its significantly lower computational demands compared to our relaxed proposals.

## 5  Discussions

In this paper, we provide a stronger classification performance on hyperbolic support vector machine using semidefinite and sparse moment-sum-of-squares relaxations on the hyperbolic support vector machine problem compared to projected gradient descent. In the one-vs-rest settings, we observe that they achieve better classification accuracy and F1 score than the existing PGD approach on both simulated and real dataset. Additionally, we discover small optimality gaps for moment-sum-of-squares relaxation, which approximately certifies global optimality of the moment solutions.

Perhaps the most critical drawback of SDP and sparse moment-sum-of-squares relaxations is their limited scalability. The runtime and memory consumption grows quickly with data size and dimension. We attempted dividing this challenge into sub-tasks, such as using one-vs-one training scheme, to cut overall runtime and memory, but observe a smaller performance advantage over the PGD baseline as they seem to perform better in this alternative setting. Also, if given the same computational resources as moment-sum-

of-squares, we suspect that PGD may be able to perform on par with the moment-relaxation method using a rather greedy hyperparameter hunt such as specifically tunning the learning rates and regularization factors.

But it is not completely without hope. For relatively large datasets, we could develop more heuristic approaches for solving our relaxed optimization problems to achieve runtime comparable with projected gradient descent. Combining the GD dynamic with interior point iterates in a problem-dependent manner could be useful Yang et al. (2023).

It remains to show if we have performance gain in either runtime or optimality by going through the dual of the problem or by designing feature kernels that map hyperbolic features to another set of hyperbolic features (Lensink et al., 2022). Nonetheless, we believe that our work introduces a valuable perspective - applying SDP and Moment relaxations - to the geometric machine learning community.

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

# A Proofs

## A.1 Deriving Soft-Margin HSVM with polynomial constraints

This section describe the key steps to transform from Equation (6) to Equation (7) for an efficient implementation in solver as well as theoretical feasibility to derive semidefinite and moment-sum-of-squares relaxations subsequently.

By introducing the slack variable $\xi_i$ as the penalty term in Equation (6), we can rewrite Equation (6) into

$$
\begin{aligned}
\min_{\boldsymbol{w} \in \mathbb{R}^{d+1}, \xi_i} \quad & \frac{1}{2}\boldsymbol{w}^T G \boldsymbol{w} + C \sum_{i=1}^{n} \xi_i \\
\text{s.t.} \quad & \xi_i \geqslant 0, \forall i \in [n] \\
& \xi_i \geqslant \operatorname{arcsinh}(1) - \operatorname{arcsinh}(-(b^{(i)})^T \boldsymbol{w}), \forall i \in [n] \\
& \boldsymbol{w}^T G \boldsymbol{w} \geqslant 0
\end{aligned}
\tag{16}
$$

then by rearranging terms and taking sinh on both sides, it follows that

$$
-(b^{(i)})^T \boldsymbol{w} \geqslant \sinh(\operatorname{arcsinh}(1) - \xi_i) = \frac{1 - \sqrt{2}}{2} e^{\xi_i} + \frac{1 + \sqrt{2}}{2} e^{-\xi_i} =: g(\xi_i),
\tag{17}
$$

where the last equality follows from hyperbolic trig identities. To make it ready for moment-sum-of-squares relaxation, we turn the function into a polynomial constraint by taking the taylor expansion up to some odd orders (we need monotonic decreasing approximation to $g(.)$, so we need odd orders).

If taking up to the first order, we relax the problem into Equation (7) [1]. If taking up to the third order, we relax the original problem to,

$$
\begin{aligned}
\min_{\boldsymbol{w} \in \mathbb{R}^{d+1}, \xi \in \mathbb{R}^n} \quad & \frac{1}{2}\boldsymbol{w}^T G \boldsymbol{w} + C \sum_{i=1}^{n} \xi_i \quad . \\
\text{s.t.} \quad & \xi_i \geqslant 0, \forall i \in [n] \\
& -(b^{(i)})^T \boldsymbol{w} \geqslant 1 - \sqrt{2}\xi_i + \frac{\xi_i^2}{2} - \sqrt{2}\frac{\xi_i^3}{6}, \forall i \in [n] \\
& \boldsymbol{w}^T G \boldsymbol{w} \geqslant 0
\end{aligned}
\tag{18}
$$

It's worth mentioning that we expect the lower bound gets tighter as we increase the order of Taylor expansion. However, once we apply the third order Taylor expansion, the constraint is no longer quadratic, eliminating the possibility of deriving a semidefinite relaxation. Instead, we must rely on moment-sum-of-squares relaxation, potentially requiring a higher order of relaxation, which may be highly time-costly.

It is also worth noting that such a Taylor expansion is consistent with the arcsinh when we allow the curvature of the hyperbolic space to vary. Specifically, one can show that as the curvature approaches 0 (i.e. as the hyperbolic space tends to Euclidean), both the arcsinh formulation in Equation (6) and Equation (7) formulation recovers the standard Euclidean SVM problem. This further justifies the naturalness of taking such a Taylor expansion.

## A.2 Stereographic projection maps a straight line on $\mathbb{H}^2$ to an arc on Poincaré ball $\mathbb{B}^2$

Suppose $\boldsymbol{w} = [w_0, w_1, w_2]$ is a valid hyperbolic decision boundary (i.e. $\boldsymbol{w} * \boldsymbol{w} < 0$), and suppose a point on the Lorentz straight line, $x = [x_0, x_1, x_2]$ with $\boldsymbol{w} * x = 0$, is mapped to a point, $v = [v_1, v_2].1$ in Poincaré space, then we have

$$
\begin{cases}
w_0 x_0 - w_1 x_1 - w_2 x_2 = 0 \\
x_0^2 - x_1^2 - x_2^2 = 1 \\
v_1 = \frac{x_1}{1 + x_0} \\
v_2 = \frac{x_2}{1 + x_0}
\end{cases}
\tag{19}
$$

If $w > 0$, we further have

$$
\left(v_1 - \frac{w_1}{w_0}\right)^2 + \left(v_2 - \frac{w_1}{w_0}\right)^2 = \frac{w_1^2 + w_2^2}{w_0^2} - 1,
\tag{20}
$$

---

[1]note that in (Cho et al., 2019), the authors use $1 - \xi_i$ instead of $1 - \sqrt{2}\xi_i$. We consider our formulation less sensitive to outliers than the former formulation.

i.e. the straight line on Poincaré space is an arc on a circle centered at $(\frac{w_1}{w_0}, \frac{w_2}{w_0})$ with radius $\sqrt{\frac{w_1^2 + w_2^2}{w_0^2} - 1}$.

One could show that if $w_0 = 0$, then it is the "arc" of a infinitely large circle, or just a Euclidean straight line passing through the origin with normal vector $(w_1, w_2)$. With this simplification, one could plot the decision boundary on the Poincaré ball easily.

### A.3   An extension to curvature other than -1

Throughout the paper, we assume that the hyperbolic space has a constant negative curvature $-1$ without loss of generality. Our analysis and relaxation easily extend to curvature other than $-1$.

Define $c$ as the negative curvature. That is, all the discussions in the main text are about $c = 1$. To derive the prooblem formulation and associated relaxation, we can follow Table 1 in Skopek et al. (2019) and retrace the proof in the supplemental material in Chien et al. (2021), then we see that the decision function, the objective, and many constraints stay the same except the following slack variable constraint in Equation (16):

$$\xi_i \geqslant \frac{1}{c}\text{arcsinh}(1) - \frac{1}{c}\text{arcsinh}(-(b^{(i)})^T\boldsymbol{w}), \forall i \in [n], \tag{21}$$

and with some rearrangements, we have

$$-(b^{(i)})^T\boldsymbol{w} \geqslant \sinh(\text{arcsinh}(1) - c\xi_i) = \frac{1 - \sqrt{2}}{2}e^{c\xi_i} + \frac{1 + \sqrt{2}}{2}e^{-c\xi_i} =: g_c(\xi_i), \tag{22}$$

whose Taylor expansion around $\xi_i = 0$ gives

$$g_c(\xi_i) \approx 1 - \sqrt{2}c\xi_i. \tag{23}$$

Essentially, this is nothing but adding a curvature factor $c$ into the problem formulation, so that solving the SVM problem in other curvature has no additional theoretical challenges. The rest follows likewise as in Appendix A.1.

The intuition behind why only this single constraint is affected can be made as follows: SVM cares about direction of the decision boundary, and if the optimal direction is found for an embedding in one $c$, the "same" direction is also optimal for the same embedding stretched to other $C$; but since the slack variables are associated to distance measures, they need to be scaled accordingly.

## B   Solution Extraction in Relaxed Formulation

In this section, we detail the heuristic methods for extracting the linear separator $\tilde{\boldsymbol{w}}$ from the solution of the relaxed model.

### B.1   Semidefinite Relaxation

For SDP, we initially construct a set of candidates $\tilde{\boldsymbol{w}}$ derived from $(\boldsymbol{W}, \boldsymbol{w}, \xi)$. Then, among candidates in this set, we choose the one that minimizes the loss function in Equation (7).

The candidates, denoted as $\tilde{\boldsymbol{w}}$'s, include

1. **Scaled top eigendirection**: $\tilde{\boldsymbol{w}} = \sqrt{\lambda_{\max}}\boldsymbol{u}_{\max}$, where $\lambda_{\max}$ and $\boldsymbol{u}_{\max}$ are the largest eigenvalue and the eigenvector associated with the largest eigenvaue;

2. **Gaussian randomizations**: sample $\tilde{\boldsymbol{w}} \sim \mathcal{N}(\boldsymbol{w}, \boldsymbol{W} - \boldsymbol{w}\boldsymbol{w}^T)$.[2] We empirically generate 10 samples from this distributions;

---

[2]a method mentioned in slide 14 of https://web.stanford.edu/class/ee364b/lectures/sdp-relax_slides.pdf

3. **Scaled matrix columns**: if it were the case that $\boldsymbol{W} = \boldsymbol{w}\boldsymbol{w}^T$, then each column of $\boldsymbol{W}$ contains $\boldsymbol{w}$ scaled by some entry within itself. Using columns of $\boldsymbol{W}$ divided by the corresponding entry of $\boldsymbol{w}$ (e.g. divide first column by $w_0$, second column by $w_1$, and so on), we get $d+1$ many candidates $\tilde{\boldsymbol{w}} = \boldsymbol{w}$'s;

4. **Nominal solution**: $\tilde{\boldsymbol{w}} = \boldsymbol{w}$, i.e. include $\boldsymbol{w}$ itself as a candidate.

Typically the top eigendirection is selected as the best candidate.

### B.2 Moment-Sum-of-Squares Relaxation

In moment-sum-of-squares relaxation, the decision variable is the truncated multi-sequence $\boldsymbol{z}$, but we could decode the solution from the moment matrix $M_\kappa[\boldsymbol{z}]$ it generates. We are able to extract the part in TMS that corresponds to $\boldsymbol{w} = [w_0, w_1, ..., w_d]$, by reading off these entries from the moment matrix, which is already a good enough solution.

For example, in $d = 2$, $\kappa = 2$, one of the sparcity group, say $\boldsymbol{q}^{(1)}$ consists of $[\boldsymbol{w}, \xi_1]$, which has monomials generated in Equation (12). Define $\otimes$ as a binary operator between two vectors of monomials that generates another vector with monomials given by the unique combinations of the product into vectors, such that

$$(\boldsymbol{w} \otimes \boldsymbol{w})^T := [w_0^2, w_1^2, w_2^2, w_0 w_1, w_0 w_2, w_1 w_2]. \tag{24}$$

Then, monomials generated can be more succinctly expressed as

$$[\boldsymbol{q}^{(1)}]_2^T = [1, \boldsymbol{w}^T, \xi_1, (\boldsymbol{w} \otimes \boldsymbol{w})^T, (\boldsymbol{w}\xi_1)^T, \xi_1^2], \tag{25}$$

and the moment matrix can be expressed in block form as

$$M_2[\boldsymbol{z}^{(1)}] = f_{\boldsymbol{z}^{(1)}} \left( \begin{bmatrix} 1 & \boldsymbol{w}^T & \xi_1 & (\boldsymbol{w} \otimes \boldsymbol{w})^T & (\boldsymbol{w}\xi_1)^T & \xi_1^2 \\ \boldsymbol{w} & \boldsymbol{w}\boldsymbol{w}^T & \boldsymbol{w}\xi_1 & \boldsymbol{w}(\boldsymbol{w} \otimes \boldsymbol{w})^T & \boldsymbol{w}(\boldsymbol{w}\xi_1)^T & \boldsymbol{w} \otimes \xi_1^2 \\ \xi_1 & (\boldsymbol{w}\xi_1)^T & \xi_1^2 & \xi_1(\boldsymbol{w} \otimes \boldsymbol{w})^T & \xi_1(\boldsymbol{w}\xi_1)^T & \xi_1^3 \\ \boldsymbol{w} \otimes \boldsymbol{w} & (\boldsymbol{w} \otimes \boldsymbol{w})\boldsymbol{w}^T & \boldsymbol{w} \otimes \boldsymbol{w}\xi_1 & \boldsymbol{w} \otimes \boldsymbol{w}(\boldsymbol{w} \otimes \boldsymbol{w})^T & \boldsymbol{w} \otimes \boldsymbol{w}(\boldsymbol{w}\xi_1)^T & \boldsymbol{w} \otimes \boldsymbol{w}\xi_1^2 \\ \boldsymbol{w}\xi_1 & \boldsymbol{w}\xi_1\boldsymbol{w}^T & \boldsymbol{w}\xi_1^2 & \boldsymbol{w}\xi_1(\boldsymbol{w} \otimes \boldsymbol{w})^T & \boldsymbol{w}\xi_1(\boldsymbol{w}\xi_1)^T & \boldsymbol{w}\xi_1^3 \\ \xi_1^2 & \xi_1^2\boldsymbol{w}^T & \xi_1^3 & \xi_1^2(\boldsymbol{w} \otimes \boldsymbol{w})^T & \xi_1^3\boldsymbol{w}^T & \xi_1^4 \end{bmatrix} \right). \tag{26}$$

Note that the value for $\boldsymbol{w}$ (the red part) is contained close to the top left corner of the moment matrix, which provides us good linear separator $\tilde{\boldsymbol{w}}$ in this problem.

## C  On Moment Sum-of-Squares Relaxation Hierarchy

In this section, we provide necessary background on moment-sum-of-squares hierarchy. We start by considering a general Polynomial Optimization Problem (POP) and introduce the sparse version. This section borrows substantially from the course note [3].

### C.1  Polynomial Optimization and Dual Cones

Polynomial optimization problem (POP) in the most generic form can be presented as

$$\text{(POP)} \ p^* = \min_{x \in \mathbb{R}^n} p(x) \quad ,$$
$$\text{s.t.} \ h_i(x) = 0 \text{ for } i = 1, 2, ..., m$$
$$g_i(x) \geq 0 \text{ for } i = 1, 2, ..., l,$$

where $p(x)$ is our polynomial objective and $h_i(x), g_i(x)$ are our polynomial equality and inequality constraints respectively. However, in general, solving such POP to global optimality is NP-hard (Lasserre, 2001; Nie,

---

[3]Chapter 5 Moment Relaxation: https://hankyang.seas.harvard.edu/Semidefinite/Moment.html

2023). To address this challenge, we leverage methods from algebraic geometry (Blekherman et al., 2012; Nie, 2023), allowing us to approximate global solutions using convex optimization methods.

To start with, we define **sum-of-squares (SOS) polynomials** as polynomials that could be expressed as a sum of squares of some other polynomials, and we define $\Sigma[x]$ to be the collection of SOS polynomials. More formally, we have

$$p(x) \in \Sigma[x] \iff \exists q_1, q_2, ..., q_m \in \mathbb{R}[x] : p(x) = \sum_{k=1}^{m} q_k^2(x),$$

where $\mathbb{R}[x]$ denotes the polynomial ring over $\mathbb{R}$.

Next, we recall the definitions of **quadratic module** and its dual. Given a set of polynomials $\mathbf{g} = [g_1, g_2, ..., g_l]$, the quadratic module generated by $\mathbf{g}$ is defined as

$$\text{Qmodule}[\mathbf{g}] = \{\sigma_0 + \sigma_1 g_1 + ... + \sigma_l g_l \mid \sigma_i\text{'s are SOS for } i \in [l]\}$$
$$= \left\{ \sum_{i=0}^{l} \sigma_i g_i \mid \sigma_i \in \Sigma[x] \text{ for } i \in [l] \right\},$$

and its degree $2d$-truncation is defined as,

$$\text{Qmodule}[\mathbf{g}]_{2d} = \left\{ \sum_{i=0}^{l} \sigma_i g_i \mid \deg(\sigma_i g_i) \le 2d, \sigma_i \in \Sigma[x] \text{ for } i \in [l] \right\},$$

where $g_0 = 1$. It has been shown that the dual cone of $\text{Qmodule}[\mathbf{g}]_{2d}$ is exactly the convex cone defined by the PSD conditions of the localizing matrices, $\mathcal{M}[g]_{2d} = \{z \in \mathbb{R}^{s(n,2d)} | M_d[z] \succcurlyeq 0, L_{d,g_i}[z] \succcurlyeq 0 \text{ for } i \in [l]\}$, where $M_d[z] = f_z([x]_d[x]_d^\intercal)$ refers to the $d^{th}$ order moment matrix, $L_{d,g_i}[z] = f_z(g_i(x) \cdot [x]_s[x]_s^\intercal)$ refers to the $d^{th}$ order localizing matrix of $g_i$ generated by $z$, and $f_z(g_i) = \langle f_z, g_i \rangle = \langle z, \text{vec}(g_i) \rangle$ refers to the linear functional associated with $z$ applied on $g_i \in \mathbb{R}[x]_{2d}$. It is worth mentioning that the application of the linear functional $f_z$ to the symmetric polynomial matrix $g(x) \cdot [x]_s[x]_s^\intercal$ is element-wise. Formally speaking, for all $g' \in \text{Qmodule}[\mathbf{g}]_{2d}$ and for all $g \in \mathcal{M}[g]_{2d}$, we have $\langle g', g \rangle \ge 0$.

Similarly, given a set of polynomials $\mathbf{h} = [h_1, h_2, ..., h_m]$, the ideal generated by $\mathbf{h}$ is defined as,

$$\text{Ideal}[\mathbf{h}] = \left\{ \sum_{i=1}^{m} \lambda_i h_i \mid \lambda_i \in \mathbb{R}[x] \text{ for } i \in [l] \right\},$$

and its degree $2d$-truncation is defined as,

$$\text{Ideal}[\mathbf{h}]_{2d} = \left\{ \sum_{i=1}^{m} \lambda_i h_i \mid \lambda_i \in \mathbb{R}[x], \deg(\lambda_i g_i) \le 2d \text{ for } i \in [l] \right\},$$

where $\lambda_i$'s are also called polynomial multipliers. Interestingly, it is shown that we can perfectly characterize the dual of the sum of ideal and quadratic module,

$$(\text{Ideal}[\mathbf{h}]_{2d} + \text{Qmodule}[\mathbf{g}]_{2d})^* = \mathcal{Z}[h]_{2d} \cap \mathcal{M}[g]_{2d},$$

where $\mathcal{Z}[h]_{2d} = \{z \in \mathbb{R}^{s(n,2d)} | L_{d,h_i}[z] = 0 \text{ for } i \in [l]\}$ is the linear subspace that linear functionals vanish on $\text{Ideal}[\mathbf{h}]_{2d}$ and $\mathcal{M}[g]_{2d} = \{z \in \mathbb{R}^{s(n,2d)} | M_d[z] \succcurlyeq 0, L_{d,g_i}[z] \succcurlyeq 0 \text{ for } i \in [l]\}$ is the convex cone defined by the PSD conditions of the localizing matrices.

With these notions setup, we can reformulate the POP above into the following SOS program for arbitrary $\kappa \in \mathbb{N}$ as the relaxataion order,

$$\gamma_\kappa^* = \max \quad \gamma$$
$$\text{s.t.} \quad p(x) - \gamma \in \text{Ideal}[\mathbf{h}]_{2\kappa} + \text{Qmodule}[\mathbf{g}]_{2\kappa},$$

whose optimal value produces a lower bound to $p^*$, i.e. $\gamma_\kappa^* \leq p^*$, and its dual problem of the SOS program above is,

$$\beta_\kappa^* = \min_{z \in \mathbb{R}^{s(n,2d)}} \langle f_z, p \rangle \quad .$$
$$\text{s.t.} \quad y \in \mathcal{Z}[h]_{2d} \cap \mathcal{M}[g]_{2d}$$
$$z_1 = 1$$

This pair of SOS programs is called the **moment-sum-of-squares hierarchy** first proposed in Lasserre (2001). It is particularly useful as it has been shown that

$$\gamma_\kappa^* \leq \beta_\kappa^* \leq p^*, \quad \text{for all } \kappa \in \mathbb{N},$$

and $\{\gamma_\kappa^*\}_\kappa$ and $\{\beta\kappa^*\}_\kappa$ are two monotonically increasing sequences. In our work, we implement our SOS programs following the dual route.

### C.2 Sparse Polynomial Optimization

In this section, we briefly discuss how sparse moment-sum-of-squares is formulated. Using the same sparsity pattern defined in Section 3 (i.e. $\boldsymbol{q}^{(i)} = (\boldsymbol{w}, \xi_i)$), we first introduce the notion of **correlated sparsity**.

**Definition 1.** *Correlated Sparsity for an objective $p \in \mathbb{R}[x]$ and associated set of constraints means*

1. *For any constraint $g_i(\boldsymbol{q}), \forall i \in [l]$, it only involves term in one sparsity group $\boldsymbol{q}^{(i)}$ for some $i \in [n]$*

2. *The objective can be split into*

$$p(\boldsymbol{q}) = \sum_{i=1}^{n} p_i(\boldsymbol{q}^{(i)}), \quad \text{for } p_i \in \mathbb{R}[\boldsymbol{q}^{(i)}], \forall i \in [n]$$

3. *The grouping satisfies the running intesection property (RIP), i.e. for all $i \in \{1, 2, ..., n-1\}$, we have*

$$\left( \cup_{k=1}^{i} \boldsymbol{q}^{(i)} \right) \cap \boldsymbol{q}^{(i+1)} \subset \boldsymbol{q}^{(s)}, \quad \text{for some } s \leqslant i$$

In our case, the first property is straightforward. For the second, we may define explicitly $p_i(\boldsymbol{q}^{(i)}) = p_i(\boldsymbol{w}, \xi_i) = \frac{1}{2n}\boldsymbol{w}^T G \boldsymbol{w} + C\xi_i$ so that we get back the original objective after summation. The last property direct follows from the star-shaped structure, i.e. $\forall i \in \{1, 2, ..., n-1\}$, we indeed have $\left( \cup_{k=1}^{i} \boldsymbol{q}^{(i)} \right) \cap \boldsymbol{q}^{(i+1)} = \boldsymbol{w} \subset \boldsymbol{q}^{(1)}$. Hence, our sparsity group indeed satisfies all three property and thus we have correlated sparsity in the problem.

With correlated sparsity and data regularity (Putinar's Positivestellentz outlined in Nie (2023)), we are able to decompose the Qmodule generated by the entire set of decision variables into the Minkowski sum of Qmodules generated by each sparsity group of variables, effectively reducing the number of decision variables in the implementations. For a problem with only inequality constraints, which is our case for HSVM, the sparse POP for our problem reads as

$$\max \ \gamma \quad ,$$
$$\text{s.t. } p(x) - \gamma \in \sum_{i=1}^{n} \text{Qmodule}[g(\boldsymbol{q}^i)]_{2\kappa}$$

and we could derive its dual accordingly and present the SDP form for implementation in Equation (14).

# D  Platt Scaling (Platt et al., 1999)

Platt scaling (Platt et al., 1999) is a common way to calibrate binary predictions to probabilistic predictions in order to generalize binary classification to multiclass classification, which has been widely used along with SVM. The key idea is that once a separator has been trained, an additional logistic regression is fitted on scores of the predictions, which can be interpreted as the closeness to the decision boundary.

In the context of HSVM, suppose $\boldsymbol{w}^*$ is the linear separator identified by the solver, then we find two scalars, $A, B \in \mathbb{R}$, with

$$P(y_i = 1|\boldsymbol{x}_i) = \frac{1}{1 + \exp\{A(\boldsymbol{w}^* * \boldsymbol{x}_i) + B\}} \tag{27}$$

where $*$ refers to the Minkowski product defined in Equation (1). The value of $A$ and $B$ are trained on the trained set using logistic regression with some additional empirical smoothing. For one-vs-rest training, we will then have $K$ sets of $(A, B)$ to train, and at the end we classify a sample to the class with the highest probability. See detailed implementation here https://home.work.caltech.edu/ htlin/program/libsvm/doc/-platt.py in `LIBSVM`.

# E  Detailed Experimental Results

This section documents the experiment details. The code base is adapted partly from `LIBSVM` [4] with a BSD-3-Clause license for Platt scaling, `hyplinear` [5] with an MIT license for PGD implementation, and `stable-hyperbolic` [6] with an MIT license for hyperbolic related functions and obtaining tree embeddings. The data used is described in Section 4.

Our `Python` code also uses some common publicly available packages, including `NumPy` (Harris et al., 2020) with a BSD license, `Matplotlib` (Hunter, 2007) with a BSD license, `Pandas` (McKinney et al., 2010) under a BSD license, `scikit-learn` (Pedregosa et al., 2011) with a BSD license, `MOSEK` (ApS, 2022) a closed-source commercial solver, and `toml` with an MIT license.

## E.1  Visualizing Decision Boundaries

Here we visualize the decision boundary of for PGD, SDP relaxation and sparse moment-sum-of-squares relaxation (Moment) on one fold of the training to provide qualitative judgements.

We first visualize training on the first fold for Gaussian 1 dataset from Figure 3 in Figure 5. We mark the train set with circles and test set with triangles, and color the decision boundary obtained by three methods with different colors. In this case, note that SDP and Moment overlap and give identical decision boundary up to machine precision, but they are different from the decision boundary of PGD method. This slight visual difference causes the performance difference displayed in Table 1.

We next visualize the decision boundary for tree 2 from Figure 3 in Figure 6. Here the difference is dramatic: we visualize both the entire data in the left panel and the zoomed-in one on the right. We indeed observe that the decision boundary from moment-sum-of-squares relaxation have roughly equal distance from points to the grey class and to the green class, while SDP relaxation is suboptimal in that regard but still enclosing the entire grey region. PGD, however, converges to a very poor local minimum that has a very small radius enclosing no data and thus would simply classify all data sample to the same class, since all data falls to one side of the decision boundary. As commented in Section 4, data imbalance is to blame, in which case the final converged solution is very sensitive to the choice of initialization and other hyperparameters such as learning rate. This is in stark contrast with solving problems using the interior point method, where after implementing into `MOSEK`, we are essentially care-free. From this example, we see that empirically sparse moment-sum-of-squares relaxation finds linear separator of the best quality, particularly in cases where PGD is expected to fail.

---

[4] https://github.com/cjlin1/libsvm
[5] https://github.com/hhcho/hyplinear
[6] https://github.com/yangshengaa/stable-hyperbolic

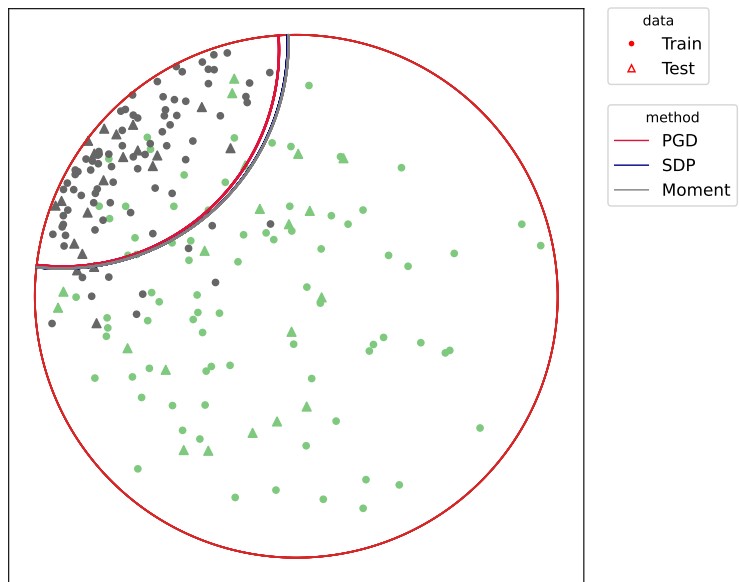

Figure 5: Decision boundary obtained by each method on one fold of train test split on Gaussian 1 dataset in Figure 3. While SDP and moment overlap, they differ from the PGD solution.

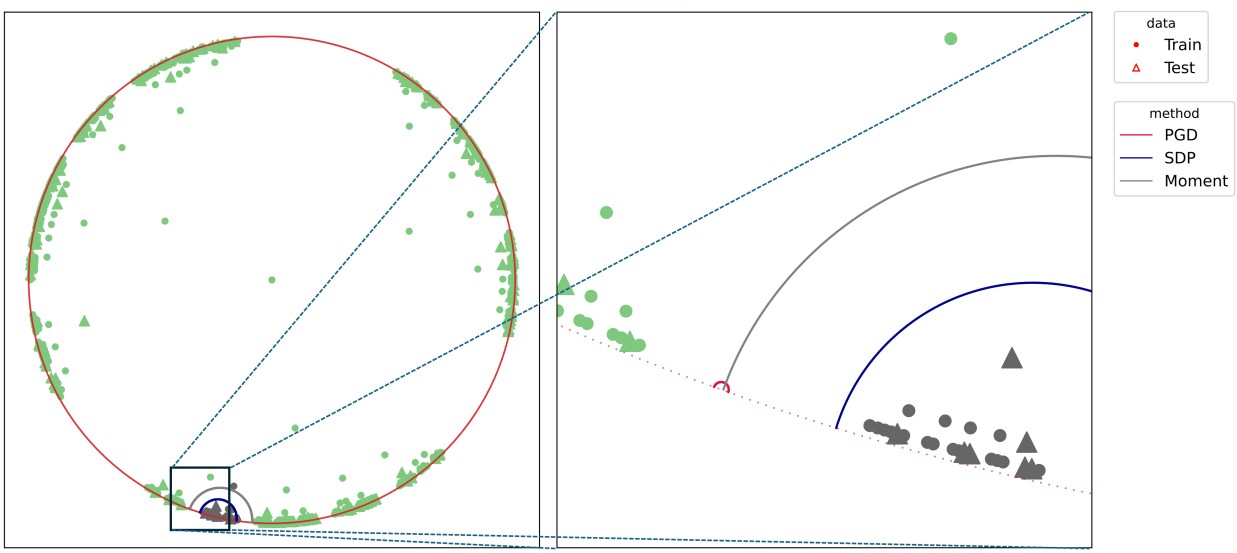

Figure 6: Decision boundary visualization of the train test split from the first fold. The left panel shows all the data and the right panel zooms in to the decision boundary. PGD gets stuck in a bad local minima (a tiny circle in the right panel) and thus classify all data samples to one class. While both SDP and moment relaxation give a decision boundary that demarcate one class from another, Moment has roughly equal margin to samples from the grey class and to samples from the green class, which is preferred in large-margin learning.

### E.2 Synthetic Gaussian

To generate mixture of Gaussian in hyperbolic space, we first generate them in Euclidean space, with the center coordinates independently drawn from a standard normal distribution. $K$ such centers are drawn for defining $K$ different classes. Then we sample isotropic Gaussian at respective center with scale $s$. Finally, we lift the generated Gaussian mixtures to hyperbolic spaces using $\exp_0$. For simplicity, we only present results for the extreme values: $K \in \{2, 5\}$, $s \in \{0.4, 1\}$, and $C \in \{0.1, 10\}$.

For each method (PGD, SDP, Moment), we compute the train/test accuracy, weighted F1 score, and loss on each of the 5 folds of data for a specific $(K, s, C)$ configuration. We then average these metrics across the 5 folds, for all methods and configurations. To illustrate the performance, we plot the improvements of the average metrics of the Moment and SDP methods compared to PGD as bar plots for 15 different seeds. Outliers beyond the interquartile range (Q1 and Q3) are excluded for clarity, and a zero horizontal line is marked for reference. Additionally, to compare the Moment and SDP methods, we compute the average optimality gaps similarly, defined in Equation (15), and present them as bar plots. Our analysis begins by examining the train/test accuracy and weighted F1 score of the PGD, SDP, and Moment methods across various synthetic Gaussian configurations, as shown in Figures 7 to 10.

Across various configurations, we observe that both the Moment and SDP methods generally show improvements over PGD in terms of train and test accuracy as well as weighted F1 score. Notably, we observe that Moment method often shows more consistent improvements compared to SDP. This consistency is evident across different values of $(K, s, C)$, suggesting that the Moment method is more robust and provide more generalizable decision boundaries. Moreover, we observe that 1. for larger number of classes (i.e. larger $K$), the Moment method consistently and significantly outperforms both SDP and PGD, highlighting its capability to manage complex class structures efficiently; and 2. for simpler datasets (with smaller scale $s$), both Moment and SDP methods generally outperform PGD, where the Moment method particularly shows a promising performance advantage over both PGD and SDP.

Next, we move to examine the train/test loss improvements compared to PGD and optimality gaps comparison across various configurations, shown in Figures 11 to 14. We observe that for $K = 5$, the Moment method achieves significantly smaller losses compared to both PGD and SDP, which aligns with our previous observations on accuracy and weighted F1 scores. However, for $K = 2$, the losses of the Moment and SDP methods are generally larger than PGD's. Nevertheless, it is important to note that these losses are not direct measurements of our optimization methods' quality; rather, they measure the quality of the extracted solutions. Therefore, a larger loss does not necessarily imply that our optimization methods are inferior to PGD, as the heuristic extraction methods might significantly impact the loss. Additionally, we observe that the optimality gaps of the Moment method are significantly smaller than those of the SDP method, suggesting that Moment provides better solutions. Interestingly, the optimality gaps of the Moment method also exhibit smaller variance compared to SDP, as indicated by the smaller boxes in the box plots, further supporting the consistency and robustness of the Moment method.

Table 5: Average runtime to finish 1 fold of training for each model on synthetic dataset.

| data | runtime | | |
|---|---|---|---|
| | **PGD** | **SDP** | **Moment** |
| gaussian 1 | 0.99s | 0.52s | 6.60s |
| gaussian 2 | 2.83s | 0.56s | 30.59s |
| gaussian 3 | 4.19s | 0.76s | 51.84s |
| tree 1 | 2.17s | 0.95s | 39.89s |
| tree 2 | 2.16s | 0.92s | 51.18s |
| tree 3 | 1.67s | 0.74s | 59.68s |

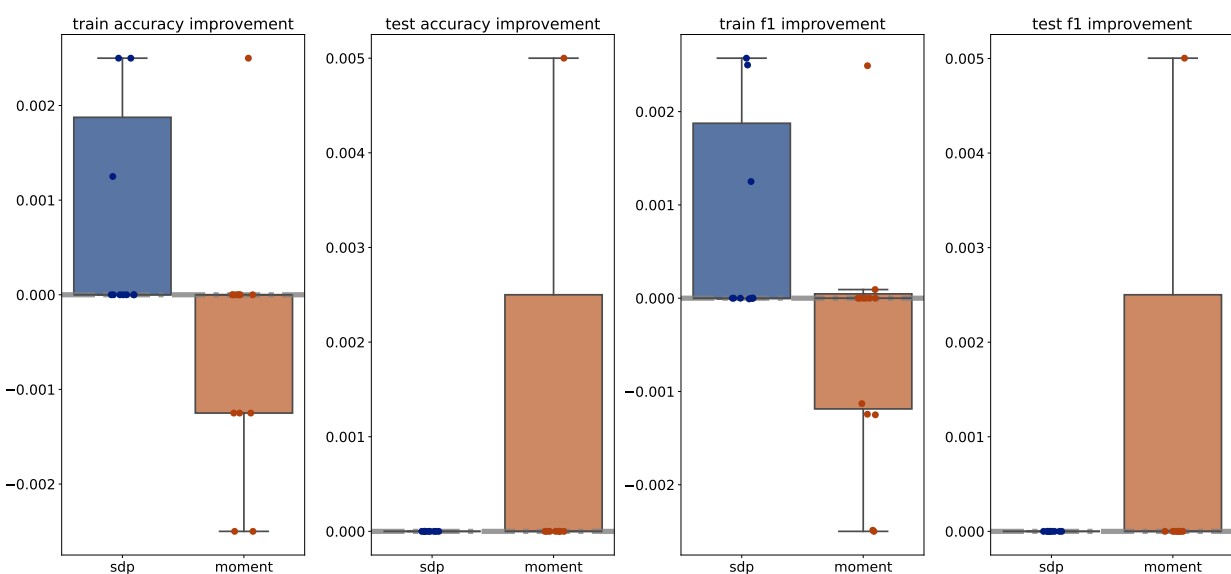

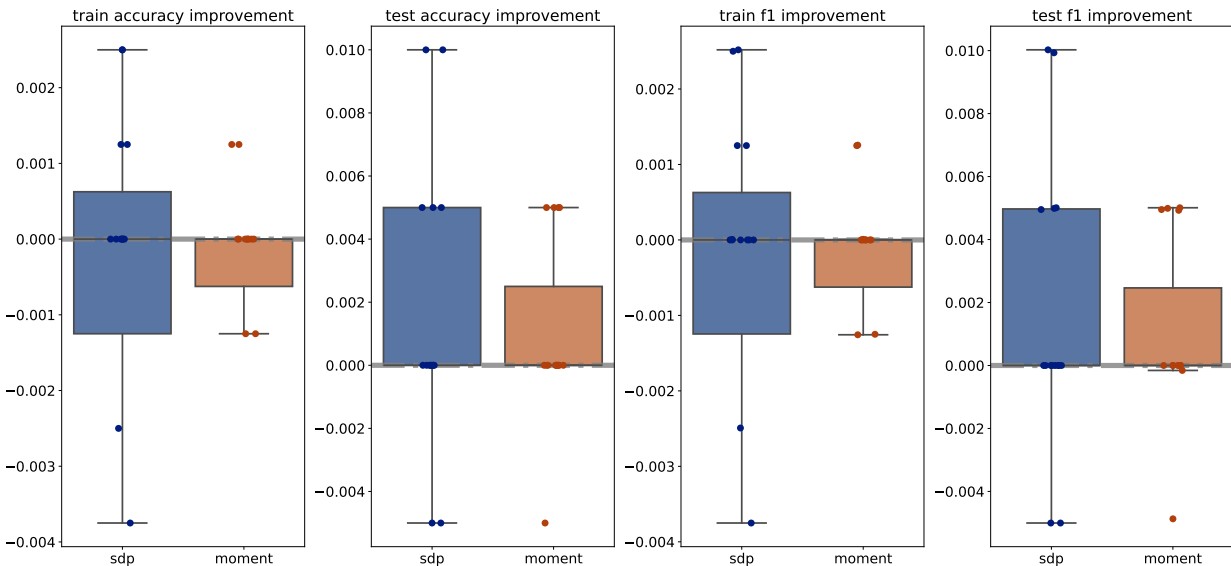

Figure 7: Train/test accuracy and train/test f1 improvements compared to PGD across various $C \in \{0.1, 10\}$ for $K = 2$ and $s = 0.4$

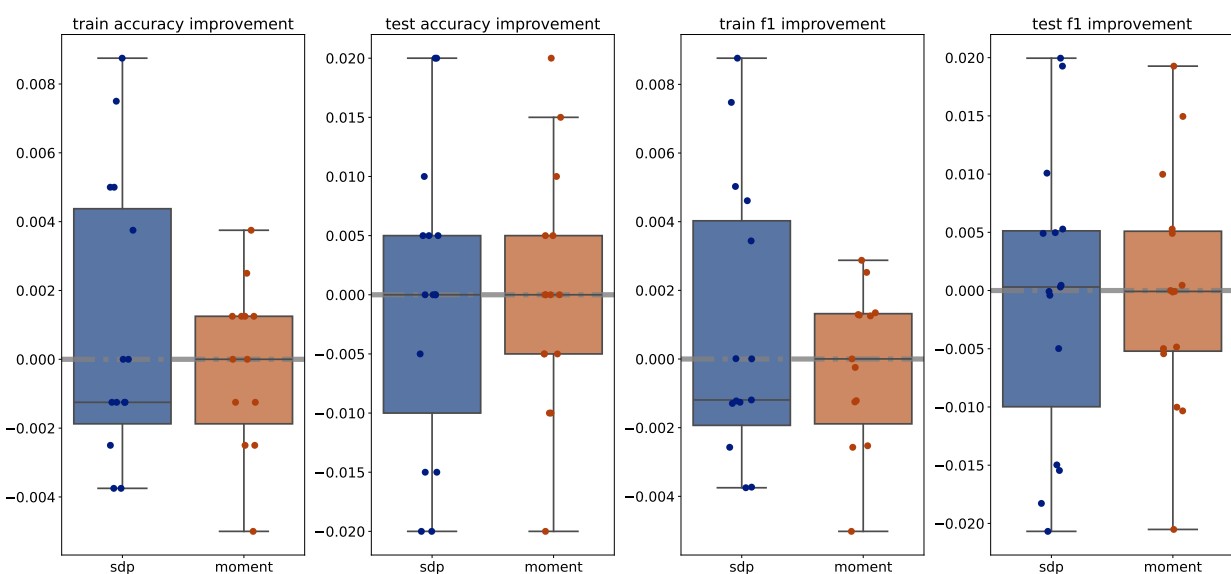

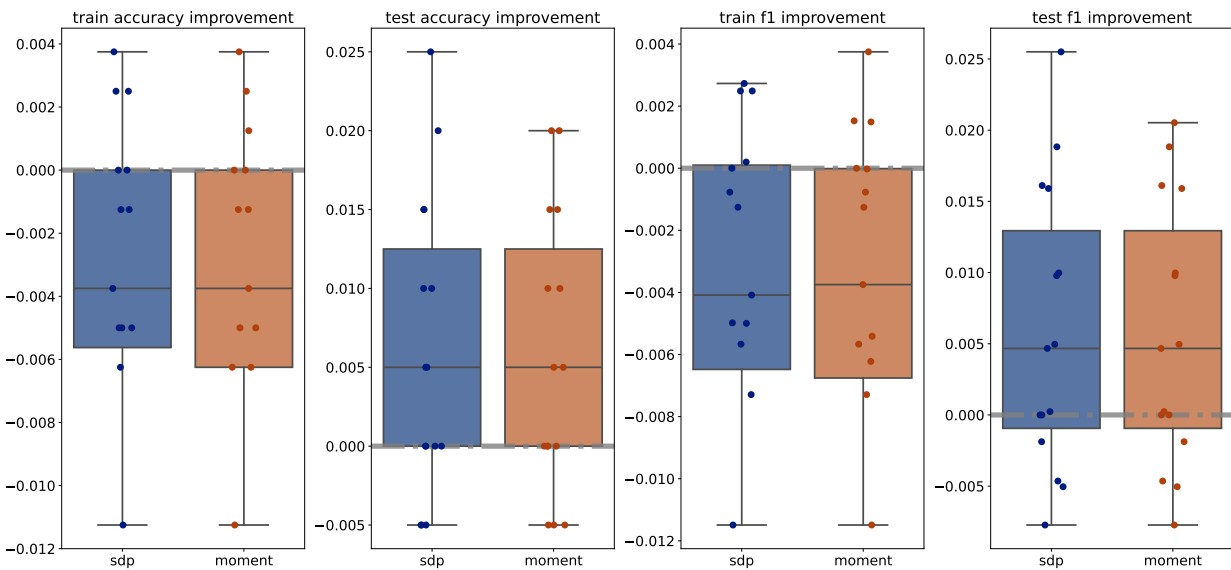

Figure 8: Train/test accuracy and train/test f1 improvements compared to PGD across various $C \in \{0.1, 10\}$ and $C$ for $K = 2$ and $s = 1.0$

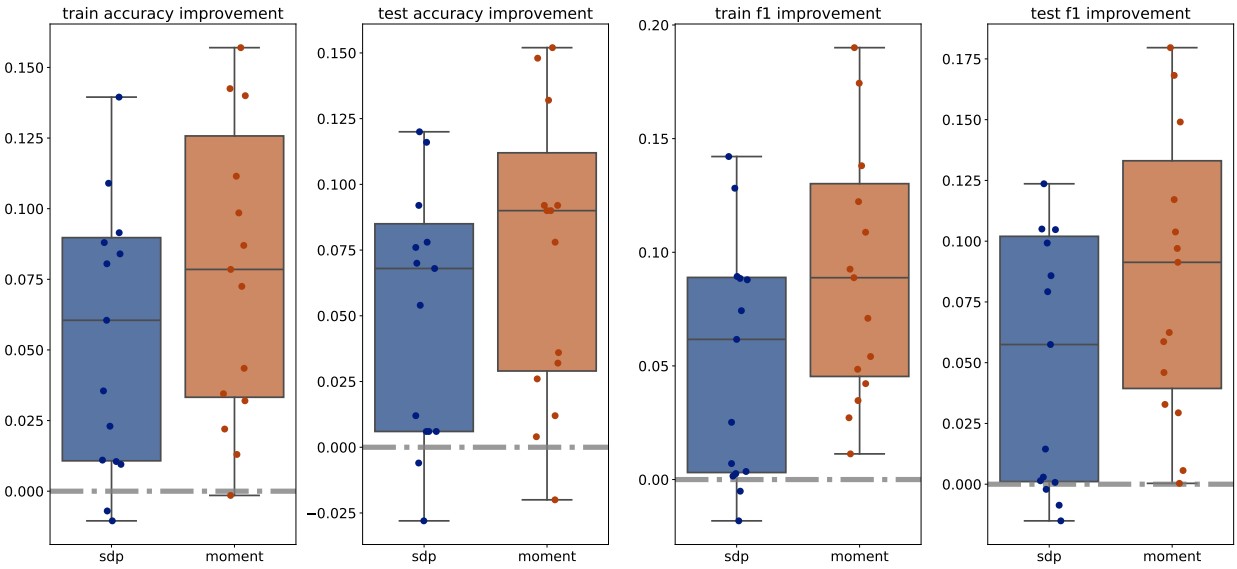

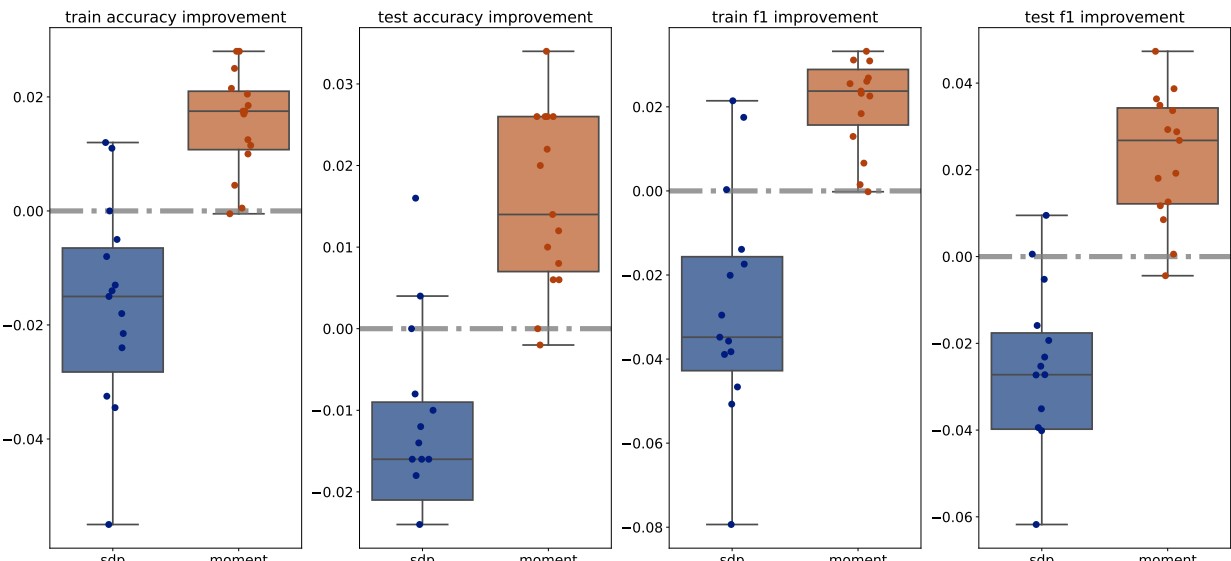

Figure 9: Train/test accuracy and train/test f1 improvements compared to PGD across various $C \in \{0.1, 10\}$ for $K = 5$ and $s = 0.4$

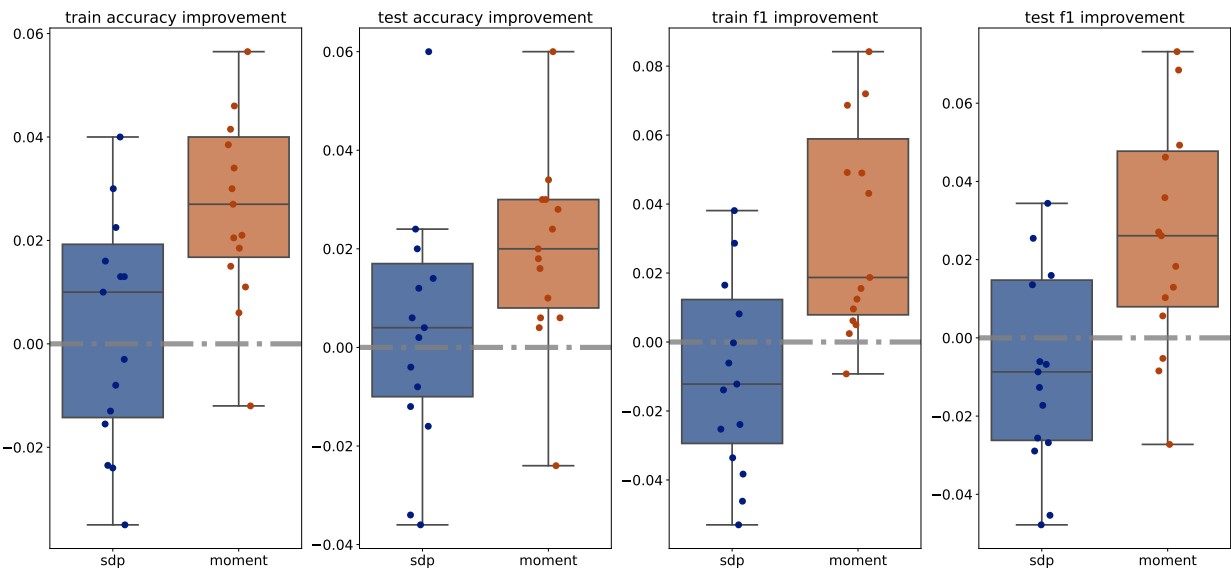

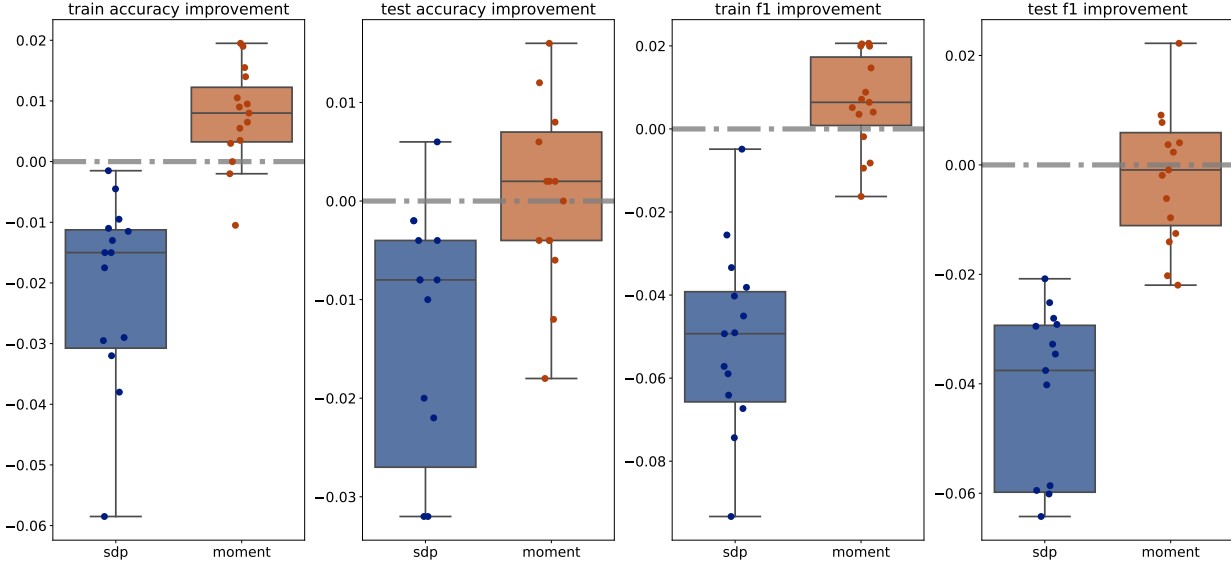

Figure 10: Train/test accuracy and train/test f1 improvements compared to PGD across various $C \in \{0.1, 10\}$ for $K = 2$ and $s = 1.0$

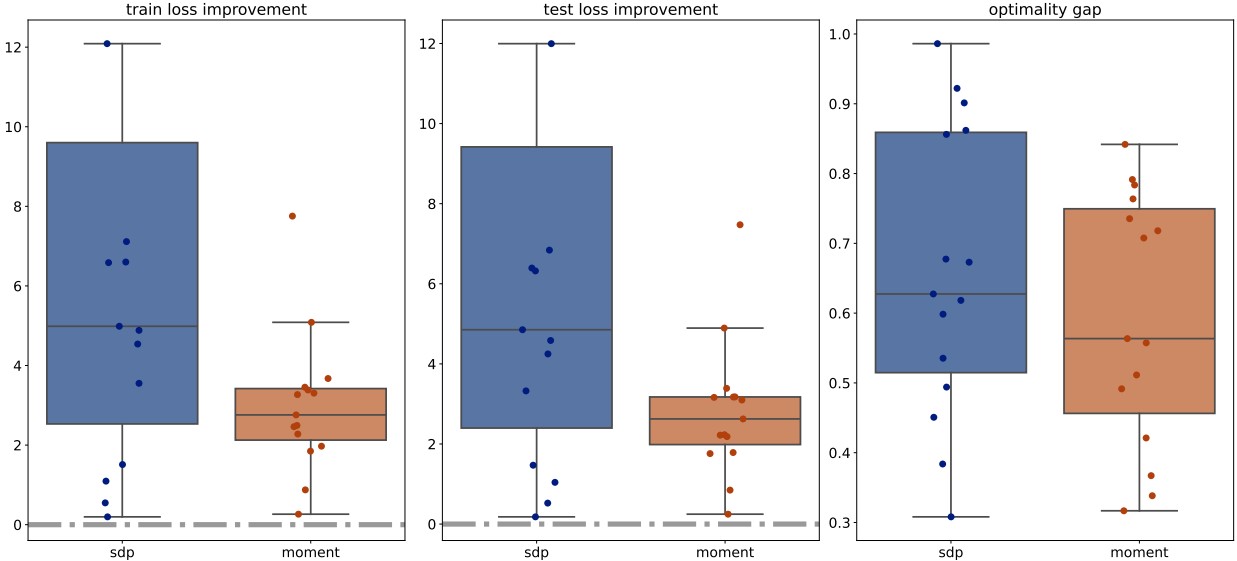

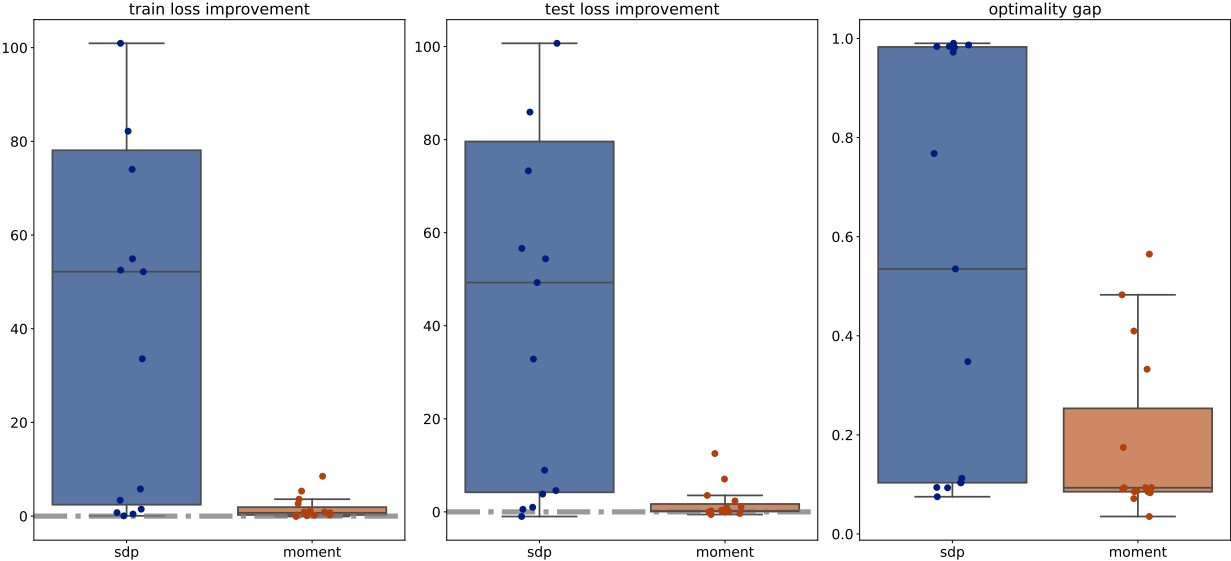

Figure 11: Train/test loss improvements compared to PGD and optimality gaps comparison across various $C \in \{0.1, 10\}$ for $K = 2$ and $s = 0.4$

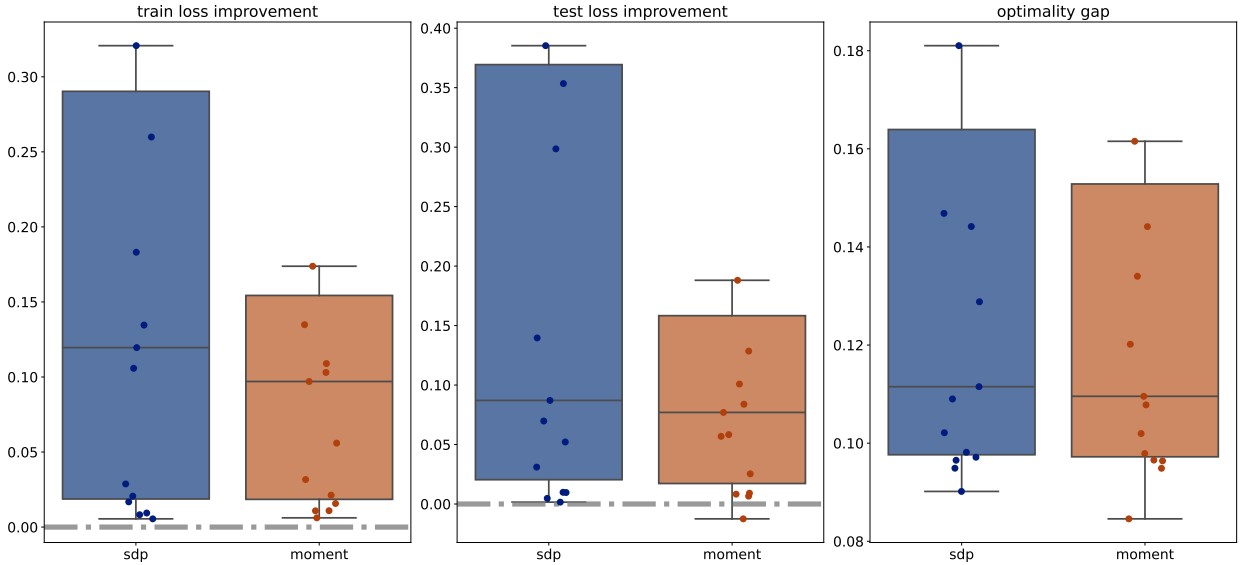

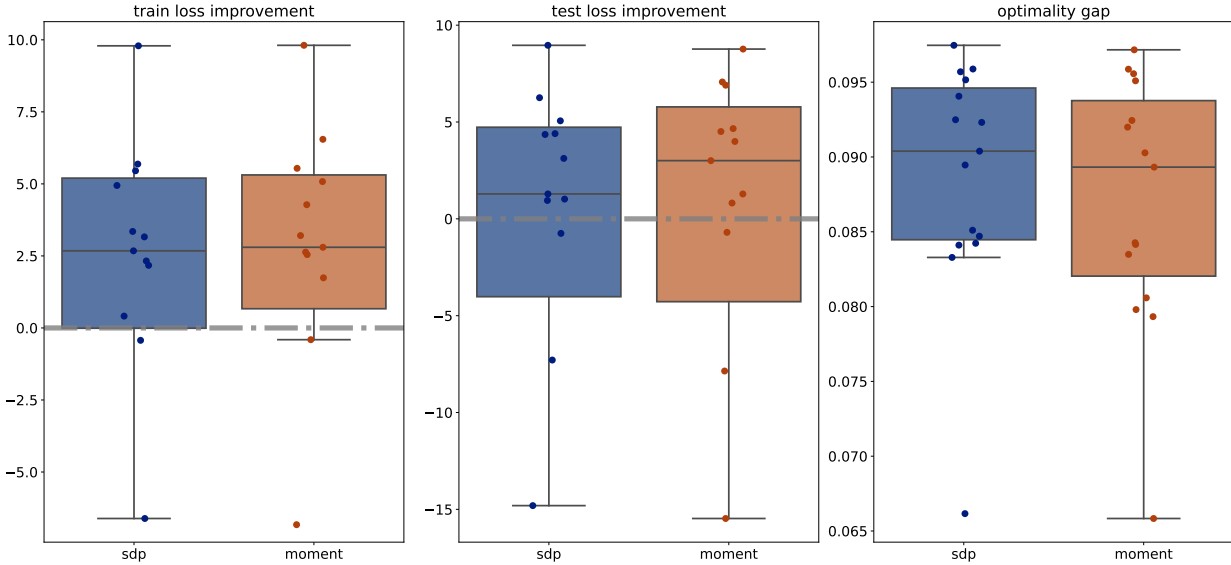

Figure 12: Train/test loss improvements compared to PGD and optimality gaps comparison across various $C \in \{0.1, 10\}$ for $K = 2$ and $s = 1$

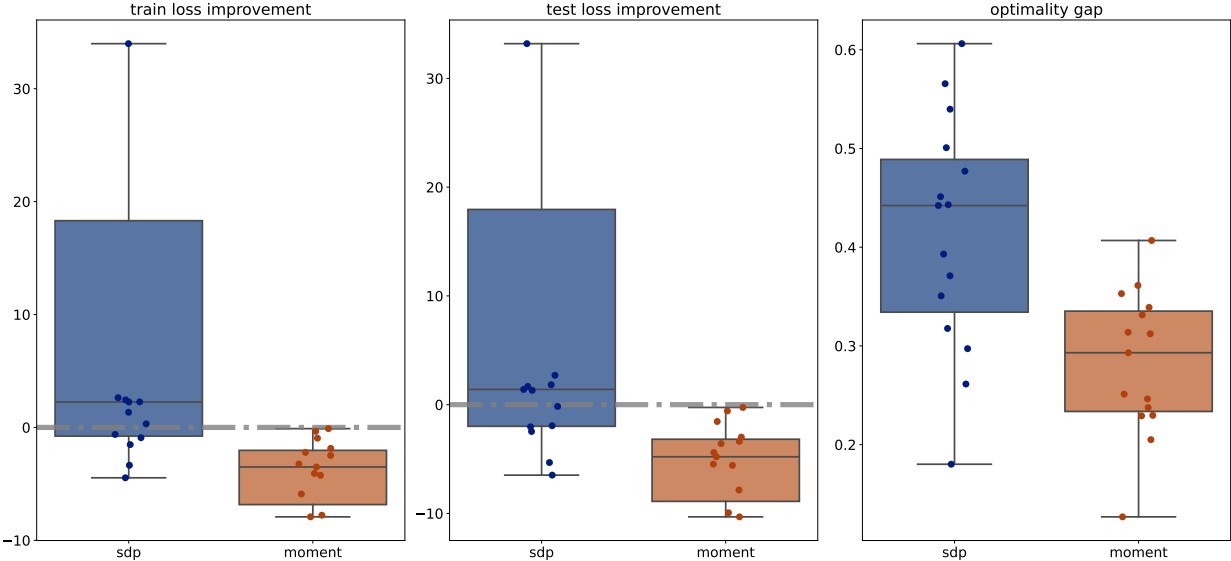

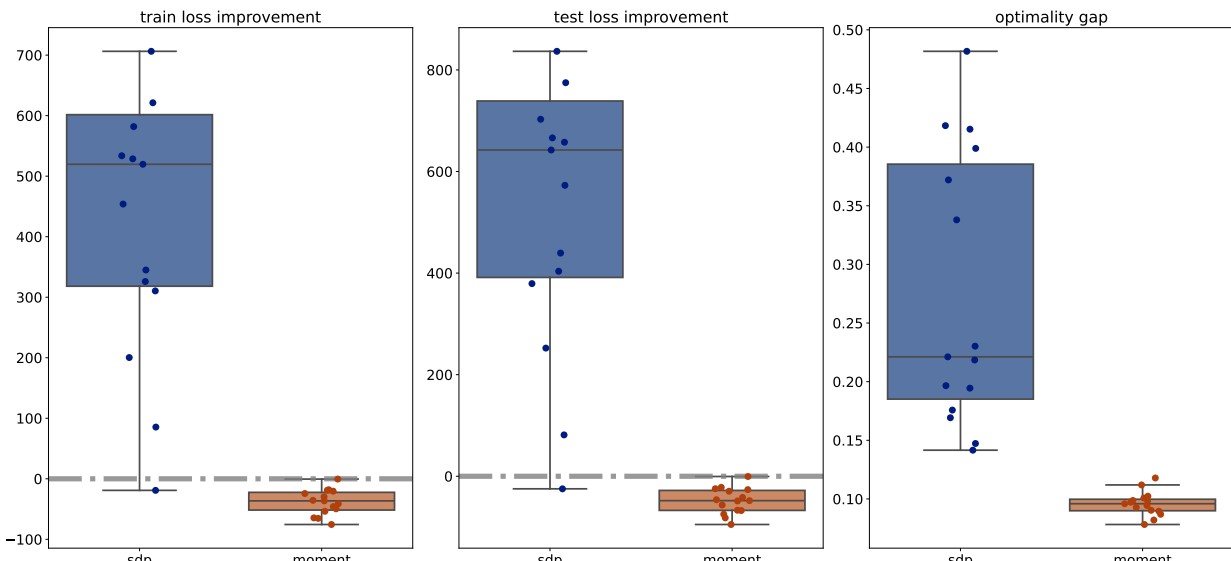

Figure 13: Train/test loss improvements compared to PGD and optimality gaps comparison across various $C \in \{0.1, 10\}$ for $K = 5$ and $s = 0.4$

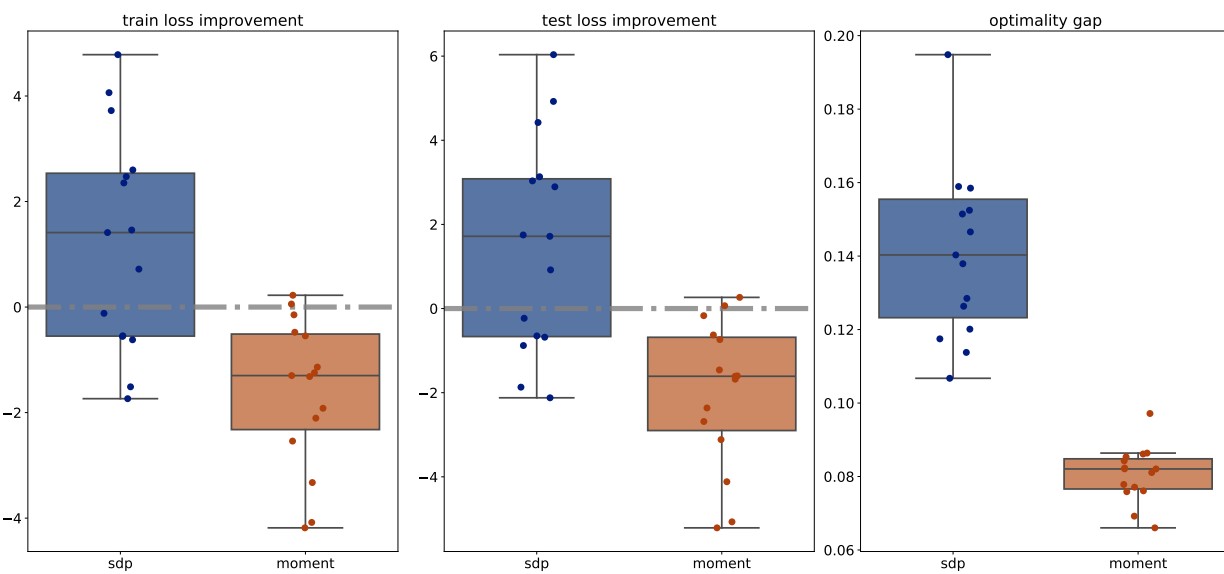

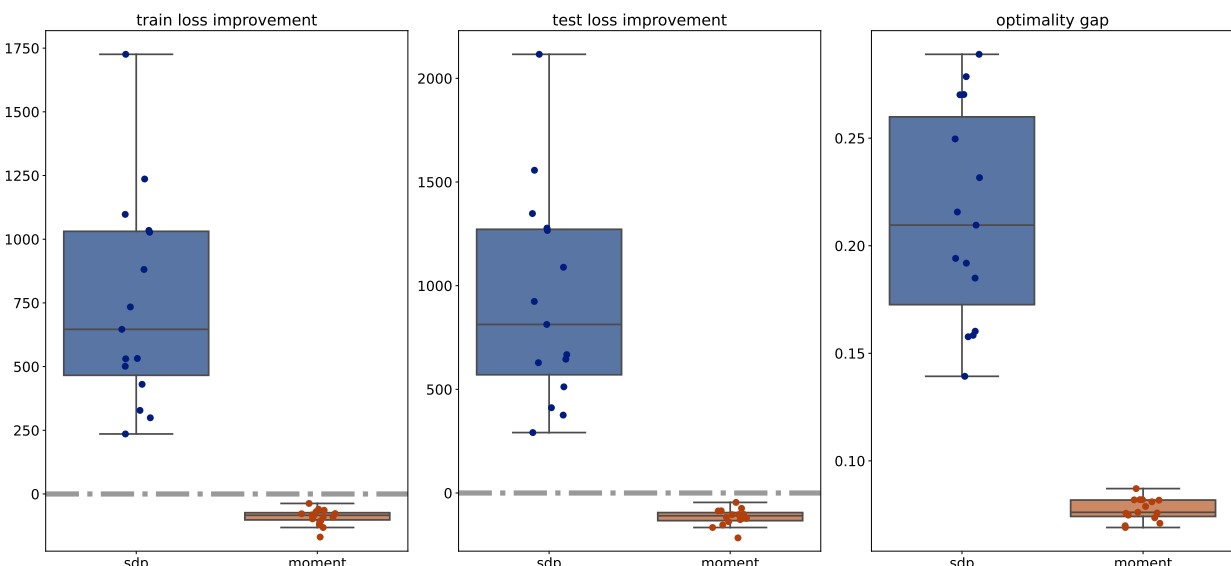

Figure 14: Train/test loss improvements compared to PGD and optimality gaps comparison across various $C \in \{0.1, 10\}$ for $K = 5$ and $s = 1$

Lastly, we compare the computational efficiency of these methods, where we compute the average runtime to finish 1 fold of training for each model on synthetic dataset, shown in Table 5. We observe that sparse moment relaxation typically requires at least one order of magnitude in runtime compared to other methods, which to some extent limits the applicability of this method to large scale dataset.

### E.3   Real Data

In this section we provide detailed performance breakdown by the choice of regularization $C$ for both one-vs-one and one-vs-rest scheme in Tables 6 to 11.

Table 6: Real dataset performance ($C = 0.1$), one-vs-rest

| data | test acc | | | test f1 (micro) | | | $\eta$ | |
| --- | --- | --- | --- | --- | --- | --- | --- | --- |
| | PGD | SDP | Moment | PGD | SDP | Moment | SDP | Moment |
| football | **40.00% ± 5.07%** | 31.30% ± 1.74% | 37.39% ± 4.43% | **0.28 ± 0.05** | 0.21 ± 0.01 | 0.26 ± 0.03 | 0.3928 | **0.1145** |
| karate | 50.00% ± 6.39% | 50.00% ± 6.39% | 50.00% ± 6.39% | 0.34 ± 0.07 | 0.34 ± 0.07 | 0.34 ± 0.07 | 0.9155 | **0.0818** |
| polbooks | 83.81% ± 2.33% | **84.76% ± 1.90%** | **84.76% ± 1.90%** | 0.79 ± 0.02 | **0.80 ± 0.03** | **0.80 ± 0.03** | 0.6334 | **0.3908** |
| krumsiek | 67.24% ± 2.64% | 82.56% ± 2.01% | **85.69% ± 0.85%** | 0.65 ± 0.02 | 0.80 ± 0.03 | **0.83 ± 0.01** | 0.6844 | **0.2025** |
| moignard | 58.13% ± 2.31% | 63.50% ± 1.96% | **63.60% ± 1.66%** | 0.52 ± 0.03 | **0.60 ± 0.02** | **0.60 ± 0.02** | **0.0435** | 0.0482 |
| olsson | 54.23% ± 0.62% | 78.99% ± 1.31% | **81.51% ± 3.32%** | 0.43 ± 0.01 | 0.76 ± 0.02 | **0.79 ± 0.04** | 0.6734 | **0.3418** |
| paul | 32.03% ± 1.23% | 53.72% ± 2.42% | **64.53% ± 2.47%** | 0.22 ± 0.02 | 0.47 ± 0.03 | **0.60 ± 0.03** | 0.4477 | **0.1214** |
| myeloidprogenitors | 50.39% ± 2.75% | 69.65% ± 4.73% | **76.69% ± 2.31%** | 0.42 ± 0.03 | 0.65 ± 0.05 | **0.75 ± 0.02** | 0.6320 | **0.3018** |

Table 7: Real dataset performance ($C = 1.0$), one-vs-rest

| data | test acc | | | test f1 (micro) | | | $\eta$ | |
| --- | --- | --- | --- | --- | --- | --- | --- | --- |
| | PGD | SDP | Moment | PGD | SDP | Moment | SDP | Moment |
| football | **40.87% ± 4.43%** | 32.17% ± 4.43% | 37.39% ± 4.43% | **0.29 ± 0.03** | 0.23 ± 0.04 | 0.26 ± 0.03 | 0.3430 | **0.1196** |
| karate | 50.00% ± 6.39% | 50.00% ± 6.39% | 50.00% ± 6.39% | 0.34 ± 0.07 | 0.34 ± 0.07 | 0.34 ± 0.07 | 0.9789 | **0.0816** |
| polbooks | 84.76% ± 1.90% | 84.76% ± 1.90% | 84.76% ± 1.90% | 0.79 ± 0.02 | **0.80 ± 0.03** | **0.80 ± 0.03** | 0.3828 | **0.1685** |
| krumsiek | 78.19% ± 1.83% | 81.55% ± 1.13% | **86.16% ± 0.81%** | 0.74 ± 0.02 | 0.78 ± 0.02 | **0.84 ± 0.01** | 0.7520 | **0.1014** |
| moignard | 63.37% ± 0.70% | 63.68% ± 1.75% | **63.78% ± 1.57%** | **0.62 ± 0.01** | 0.60 ± 0.02 | 0.60 ± 0.02 | **0.0299** | 0.0401 |
| olsson | 58.31% ± 0.64% | 79.63% ± 3.54% | **81.20% ± 3.68%** | 0.48 ± 0.00 | 0.77 ± 0.04 | **0.79 ± 0.04** | 0.5281 | **0.0976** |
| paul | 54.86% ± 1.26% | 48.66% ± 3.69% | **64.64% ± 2.37%** | 0.48 ± 0.02 | 0.41 ± 0.05 | **0.61 ± 0.02** | 0.4053 | **0.0936** |
| myeloidprogenitors | 59.94% ± 0.96% | 70.12% ± 3.28% | **76.84% ± 2.04%** | 0.54 ± 0.02 | 0.67 ± 0.04 | **0.75 ± 0.02** | 0.6570 | **0.1074** |

Table 8: Real dataset performance ($C = 10.0$), one-vs-rest

| data | test acc | | | test f1 (micro) | | | $\eta$ | |
| --- | --- | --- | --- | --- | --- | --- | --- | --- |
| | PGD | SDP | Moment | PGD | SDP | Moment | SDP | Moment |
| football | **41.74% ± 6.51%** | 34.78% ± 6.15% | 37.39% ± 4.43% | **0.29 ± 0.05** | 0.23 ± 0.07 | 0.26 ± 0.03 | 0.3130 | **0.0999** |
| karate | 50.00% ± 6.39% | 50.00% ± 6.39% | 50.00% ± 6.39% | 0.34 ± 0.07 | 0.34 ± 0.07 | 0.34 ± 0.07 | 0.9986 | **0.0921** |
| polbooks | 83.81% ± 2.33% | **84.76% ± 1.90%** | **84.76% ± 1.90%** | 0.79 ± 0.02 | **0.80 ± 0.03** | **0.80 ± 0.03** | 0.1711 | **0.0991** |
| krumsiek | 81.78% ± 2.66% | 78.66% ± 2.12% | **86.47% ± 0.64%** | 0.79 ± 0.03 | 0.74 ± 0.03 | **0.84 ± 0.00** | 0.8008 | **0.0921** |
| moignard | 60.40% ± 1.03% | 63.60% ± 1.80% | **63.78% ± 1.57%** | 0.58 ± 0.02 | **0.60 ± 0.02** | **0.60 ± 0.02** | **0.0338** | 0.0396 |
| olsson | 74.27% ± 5.15% | 79.63% ± 3.54% | **79.94% ± 4.11%** | 0.69 ± 0.07 | **0.77 ± 0.04** | 0.77 ± 0.05 | 0.4118 | **0.0659** |
| paul | 46.61% ± 1.95% | 47.16% ± 2.20% | **64.71% ± 2.36%** | 0.36 ± 0.02 | 0.37 ± 0.03 | **0.61 ± 0.02** | 0.4034 | **0.0861** |
| myeloidprogenitors | 69.34% ± 3.81% | 65.74% ± 3.58% | **76.69% ± 2.14%** | 0.66 ± 0.05 | 0.61 ± 0.04 | **0.75 ± 0.02** | 0.7076 | **0.0842** |

In one-vs-rest scheme, we observe that the Moment method consistently outperforms both PGD and SDP across almost all datasets and $C$ in terms of accuracy and F1 scores. Notably, the optimality gaps, $\eta$, for Moment are consistently lower than those for SDP, indicating that the Moment method's solution obatin a better gap, which underscore the effectiveness of the Moment method in real datasets.

Table 9: Real dataset performance ($C = 0.1$), one-vs-one

| data | test acc | | | test f1 (micro) | | | $\eta$ | |
| --- | --- | --- | --- | --- | --- | --- | --- | --- |
| | PGD | SDP | Moment | PGD | SDP | Moment | SDP | Moment |
| football | 39.13% ± 9.12% | **42.61% ± 5.07%** | 41.74% ± 7.06% | 0.31 ± 0.06 | **0.35 ± 0.06** | 0.33 ± 0.07 | 0.8495 | **0.7177** |
| karate | 50.00% ± 6.39% | 50.00% ± 6.39% | 50.00% ± 6.39% | 0.34 ± 0.07 | 0.34 ± 0.07 | 0.34 ± 0.07 | 0.9155 | **0.0818** |
| polbooks | 81.90% ± 4.67% | **86.67% ± 1.90%** | 83.81% ± 2.33% | 0.79 ± 0.05 | **0.84 ± 0.03** | 0.81 ± 0.03 | 0.9237 | **0.6320** |
| krumsiek | 89.76% ± 0.80% | **90.46% ± 1.18%** | 90.38% ± 1.49% | **0.90 ± 0.01** | **0.90 ± 0.01** | 0.90 ± 0.02 | 0.5843 | **0.3855** |
| moignard | **64.31% ± 1.13%** | 62.38% ± 1.56% | 62.35% ± 1.39% | **0.63 ± 0.01** | 0.61 ± 0.01 | 0.61 ± 0.01 | 0.0852 | 0.0576 |
| olsson | 93.41% ± 1.84% | 94.04% ± 1.21% | **94.36% ± 0.78%** | 0.93 ± 0.02 | **0.94 ± 0.01** | **0.94 ± 0.01** | 0.5102 | **0.4412** |
| paul | 64.86% ± 2.50% | **68.85% ± 2.26%** | 67.75% ± 2.26% | 0.62 ± 0.03 | **0.66 ± 0.02** | 0.65 ± 0.03 | 0.6974 | **0.5787** |
| myeloidprogenitors | 79.50% ± 2.50% | 80.44% ± 2.47% | **80.44% ± 1.84%** | 0.79 ± 0.02 | **0.80 ± 0.02** | **0.80 ± 0.02** | 0.6504 | **0.5340** |
| cifar | **98.43% ± 0.16%** | 98.42% ± 0.18% | 98.43% ± 0.18% | 0.98 ± 0.00 | 0.98 ± 0.00 | 0.98 ± 0.00 | 0.1505 | **0.0902** |
| fashion-mnist | 95.04% ± 0.21% | **95.28% ± 0.16%** | 95.12% ± 0.18% | 0.95 ± 0.00 | 0.95 ± 0.00 | 0.95 ± 0.00 | 0.4262 | **0.1555** |

Table 10: Real dataset performance ($C = 1.0$), one-vs-one

| data | test acc | | | test f1 (micro) | | | $\eta$ | |
|---|---|---|---|---|---|---|---|---|
| | PGD | SDP | Moment | PGD | SDP | Moment | SDP | Moment |
| football | 40.00% ± 6.39% | **42.61% ± 5.07%** | 41.74% ± 7.06% | 0.32 ± 0.06 | **0.35 ± 0.06** | 0.33 ± 0.07 | 0.7842 | **0.5012** |
| karate | 50.00% ± 6.39% | 50.00% ± 6.39% | 50.00% ± 6.39% | 0.34 ± 0.07 | 0.34 ± 0.07 | 0.34 ± 0.07 | 0.9789 | **0.0816** |
| polbooks | 82.86% ± 3.81% | **86.67% ± 1.90%** | 83.81% ± 2.33% | 0.79 ± 0.04 | **0.84 ± 0.03** | 0.81 ± 0.03 | 0.7263 | **0.2733** |
| krumsiek | 89.05% ± 1.56% | **90.46% ± 1.18%** | 90.22% ± 1.66% | 0.89 ± 0.02 | **0.90 ± 0.01** | 0.90 ± 0.02 | 0.8171 | **0.2330** |
| moignard | **63.22% ± 0.92%** | 62.43% ± 1.11% | 62.66% ± 1.19% | **0.62 ± 0.01** | 0.61 ± 0.01 | 0.61 ± 0.01 | **0.0402** | 0.0419 |
| olsson | 92.47% ± 2.35% | 94.03% ± 2.12% | **94.36% ± 0.78%** | 0.92 ± 0.03 | 0.94 ± 0.02 | **0.94 ± 0.01** | 0.7517 | **0.3635** |
| paul | 66.98% ± 2.89% | **68.89% ± 2.28%** | 67.97% ± 2.34% | 0.64 ± 0.03 | **0.66 ± 0.02** | 0.66 ± 0.03 | 0.7191 | **0.4195** |
| myeloidprogenitors | 80.13% ± 1.99% | 80.28% ± 2.50% | **80.44% ± 1.84%** | 0.80 ± 0.02 | 0.80 ± 0.02 | 0.80 ± 0.02 | 0.7540 | **0.3519** |
| cifar | **98.42% ± 0.16%** | 98.42% ± 0.17% | 98.42% ± 0.17% | 0.98 ± 0.00 | 0.98 ± 0.00 | 0.98 ± 0.00 | 0.0959 | **0.0573** |
| fashion-mnist | 94.97% ± 0.22% | **95.27% ± 0.16%** | 95.20% ± 0.16% | 0.95 ± 0.00 | 0.95 ± 0.00 | 0.95 ± 0.00 | 0.3635 | **0.0581** |

Table 11: Real dataset performance ($C = 10.0$), one-vs-one

| data | test acc | | | test f1 (micro) | | | $\eta$ | |
|---|---|---|---|---|---|---|---|---|
| | PGD | SDP | Moment | PGD | SDP | Moment | SDP | Moment |
| football | 40.00% ± 5.07% | **42.61% ± 5.07%** | 41.74% ± 7.06% | 0.32 ± 0.06 | **0.35 ± 0.06** | 0.33 ± 0.07 | 0.6699 | **0.2805** |
| karate | 50.00% ± 6.39% | 50.00% ± 6.39% | 50.00% ± 6.39% | 0.34 ± 0.07 | 0.34 ± 0.07 | 0.34 ± 0.07 | 0.9986 | **0.0921** |
| polbooks | 83.81% ± 3.81% | **86.67% ± 1.90%** | 83.81% ± 2.33% | 0.80 ± 0.04 | **0.84 ± 0.03** | 0.81 ± 0.03 | 0.3383 | **0.1051** |
| krumsiek | 89.52% ± 0.75% | **90.46% ± 1.18%** | 89.60% ± 1.68% | 0.89 ± 0.01 | **0.90 ± 0.01** | 0.89 ± 0.02 | 0.9211 | **0.1349** |
| moignard | **63.50% ± 1.35%** | 62.53% ± 1.10% | 62.66% ± 1.17% | **0.62 ± 0.01** | 0.61 ± 0.01 | 0.61 ± 0.01 | **0.0312** | 0.0401 |
| olsson | 93.40% ± 2.75% | 94.03% ± 2.12% | **94.36% ± 0.78%** | 0.93 ± 0.03 | 0.94 ± 0.02 | **0.94 ± 0.01** | 0.9266 | **0.2534** |
| paul | 65.45% ± 2.41% | **68.85% ± 2.26%** | 68.52% ± 2.39% | 0.63 ± 0.03 | **0.66 ± 0.02** | 0.66 ± 0.03 | 0.7863 | **0.2130** |
| myeloidprogenitors | 79.81% ± 2.00% | 80.28% ± 2.50% | **80.60% ± 2.58%** | 0.80 ± 0.02 | 0.80 ± 0.02 | **0.81 ± 0.02** | 0.8911 | **0.1960** |
| cifar | 98.38% ± 0.14% | **98.42% ± 0.17%** | **98.42% ± 0.17%** | 0.98 ± 0.00 | 0.98 ± 0.00 | 0.98 ± 0.00 | 0.0825 | **0.0550** |
| fashion-mnist | 94.42% ± 1.10% | **95.28% ± 0.16%** | 95.23% ± 0.15% | 0.94 ± 0.01 | **0.95 ± 0.00** | **0.95 ± 0.00** | 0.3492 | **0.0054** |

In one-vs-one scheme however, we observe that the SDP and Moment have comparative performances, both better than PGD. Nevertheless, the optimality gaps of SDP are still significantly larger than the Moment's, for almost all cases.

Similarly, we compare the average runtime to finish 1 fold of training for each model on these real datasets, shown in Table 12. We observe a similar trend: the sparse moment relaxation typically requires at least an order of magnitude more runtime compared to the other methods.

Table 12: Average runtime to finish 1 fold of training for each model on real dataset.

| data | one-vs-rest runtime | | | one-vs-one runtime | | |
|---|---|---|---|---|---|---|
| | PGD | SDP | Moment | PGD | SDP | Moment |
| football | 5.908s | 0.581s | 20.522s | 21.722s | 1.119s | 17.907s |
| karate | 2.437s | 0.501s | 1.124s | 2.472s | 0.525s | 1.176s |
| polbooks | 2.205s | 0.547s | 5.537s | 1.748s | 0.544s | 4.393s |
| krumsiek | 9.639s | 3.053s | 294.077s | 15.728s | 1.561s | 169.176s |
| moignard | 9.690s | 13.000s | 368.433s | 9.271s | 4.826s | 293.234s |
| olsson | 7.695s | 0.738s | 106.584s | 10.567s | 0.653s | 38.487s |
| paul | 34.452s | 22.717s | 1487.205s | 75.878s | 3.542s | 1313.008s |
| myeloidprogenitors | 6.566s | 1.170s | 112.341s | 10.769s | 1.291s | 84.402s |
| cifar | - | - | - | 237.019s | 2606.295s | 9430.741s |
| fashion-mnist | - | - | - | 285.604s | 2840.226s | 13128.220s |

# F  Robust Hyperbolic Support Vector Machine

In this section, we propose the robust version of hyperbolic support vector machine without implementation. This is different from the practice of adversarial training that searches for adversarial samples on the fly used in the machine learning community, such as Weber et al. (2020). Rather, we predefine an uncertainty structure for data features and attempt to write down the corresponding optimization formulation, which we call the robust counterpart, as described in Ben-Tal et al. (2009); Bertsimas & Hertog (2022).

Denote $\bar{\boldsymbol{x}}_i$ as the features observed, or the nominal values, the QCQP can be made robust by introducing an uncertainty set $\mathcal{U}_{\bar{\boldsymbol{x}}_i}$ which defines the maximum perturbation around $\bar{\boldsymbol{x}}_i$ that we think the true data could live in. More precisely, the formulation is now

$$\text{(QCQP-robust)} \min_{\boldsymbol{w}\in\mathbb{R}^{d+1}\xi\in\mathbb{R}^n} \frac{1}{2}\boldsymbol{w}^T G\boldsymbol{w} + C\sum_{i=1}^{n}\xi_i \quad,$$

$$\text{s.t. } \xi_i \geqslant 0, \forall i \in [n] \tag{28}$$
$$(y_i(G\boldsymbol{x}))^T \boldsymbol{w} \leqslant \sqrt{2}\xi_i - 1, \forall i \in [n], \forall \boldsymbol{x} \in \mathcal{U}_{\bar{\boldsymbol{x}}_i}$$
$$\boldsymbol{w}^T G\boldsymbol{w} \geqslant 0$$

where we add data uncertainty to each classifiability constraint (the second constraint).

However, we could not naively add Euclidean perturbations around $\bar{\boldsymbol{x}}_i$ and postulate that as our uncertainty set, since Euclidean perturbations to hyperbolic features highly likely would force it outside of the hyperbolic manifold. Instead, a natural generalization to Euclidean noise in the hyperbolic space is to add the noise in the Euclidean tangent space and subsequently 'project' them back onto the hyperbolic space, so that all samples in the uncertainty set stay on the hyperbolic manifold. This is made possible through exponential and logarithmic map.

We demonstrate our robust version of HSVM using a $l_\infty$ example. Define $U_{\bar{x}}$ the uncertainty set for data $\bar{x} \in \mathbb{H}^d$ as

$$U_{\bar{x}} = \{x \in \mathbb{H}^d \mid x = \exp_0(\log_0(\bar{x}) + z), \|z\|_\infty \leqslant \rho\}, \tag{29}$$

where $\rho \geqslant 0$ is the robust parameter, controlling the scale of perturbations we are willing to tolerate. In this case, all perturbations are done on the Euclidean tangent space at the origin. Since the geometry of the set is complicated, for small $\rho$ we may relax the uncertainty set to its first order taylor expansion given by

$$\tilde{U}_{\bar{x}} = \{x \in \mathbb{H}^d \mid x = \bar{x} + J_{\bar{x}}z, \|z\|_\infty \leqslant \rho\}, \quad J_{\bar{x}} = D\exp_0(.)|_{.=\bar{x}} \tag{30}$$

where $J_{\bar{x}}$ is the Jacobian of the exponential map evaluated at $\bar{x}$. Suppose $\bar{x} = [x_0, x_r]$, where $x_r \in \mathbb{R}^d$, $x_0 \in \mathbb{R}$ (i.e. partitioned into negative and positive definite parts), and further suppose $v = \log_0(\bar{x})$, then we could write down the Jacobian as

$$J_{\bar{x}} = \left[ \begin{array}{c} x_r^T \\ \left(\frac{\cosh(\|v\|_2)}{\|v\|_2} - \frac{\sinh(\|v\|_2)}{\|v\|_2^3}\right)vv^T + \frac{\sinh(\|v\|_2)}{\|v\|_2}I_d \end{array} \right] \in \mathbb{R}^{(d+1)\times d}.$$

Then, by adding the uncertainty set to the constraints, we have

$$(y_i(Gx))^T\boldsymbol{w} \leqslant \sqrt{2}\xi_i - 1, \forall x \in \tilde{U}_{x_i} \iff (y_i(Gx_i + GJ_{x_i}z))^T\boldsymbol{w} \leqslant \sqrt{2}\xi_i - 1, \|z\|_\infty \leqslant \rho \tag{31}$$

$$\iff (y_i(Gx_i))^T\boldsymbol{w} + \rho\|y_i(GJ_{x_i})^T\boldsymbol{w}\|_1 \leqslant \sqrt{2}\xi_i - 1, \tag{32}$$

where the last step is a rewriting into the robust counterpart (RC). We present the $l_\infty$ norm bounded robust HSVM as follows,

$$
\min_{\boldsymbol{w}\in\mathbb{R}^{d+1},\xi\in\mathbb{R}^n} \frac{1}{2}\boldsymbol{w}^T G\boldsymbol{w} + C\sum_{i=1}^n \xi_i \quad .
$$
$$
\begin{aligned}
\text{s.t.} \quad &\xi_i \geqslant 0, \forall i \in [n] \\
&(y_i(Gx_i))^T\boldsymbol{w} + \rho\|y_i(GJ_{x_i})^T\boldsymbol{w}\|_1 \leqslant \sqrt{2}\xi_i - 1, \forall i \in [n] \\
&\boldsymbol{w}^T G\boldsymbol{w} \geqslant 0
\end{aligned}
\tag{33}
$$

Note that since $y_i \in \{-1, 1\}$, we may drop the $y_i$ term in the norm and subsequently write down the SDP relaxation to this non-convex QCQP problem and solve it efficiently with

$$
(\textbf{SDP-Linf}) \min_{\substack{\boldsymbol{W}\in\mathbb{R}^{(d+1)\times(d+1)} \\ \boldsymbol{w}\in\mathbb{R}^{d+1} \\ \xi\in\mathbb{R}^n}} \frac{1}{2}\textbf{Tr}(G, \boldsymbol{W}) + C\sum_{i=1}^n \xi_i \quad .
$$
$$
\begin{aligned}
\text{s.t.} \quad &\xi_i \geqslant 0, \forall i \in [n] \\
&(y_i(Gx_i))^T\boldsymbol{w} + \rho\|(GJ_{x_i})^T\boldsymbol{w}\|_1 \leqslant \sqrt{2}\xi_i - 1, \forall i \in [n] \\
&\textbf{Tr}(G, \boldsymbol{W}) \geqslant 0 \\
&\begin{bmatrix} 1 & \boldsymbol{w}^T \\ \boldsymbol{w} & \boldsymbol{W} \end{bmatrix} \succeq 0
\end{aligned}
\tag{34}
$$

For the implementation in `MOSEK`, we linearize the $l_1$ norm term by introducing extra auxiliary variables, which we do not show here. The moment relaxation can be implemented likewise, since this is constraint-wise uncertainty and we preserve the same sparsity pattern so that the same sparse moment relaxation applies.

**Remark 1.** *Since we take a Taylor approximation to first order, $\tilde{U}_x$ also does not guarantee that all of its elements live strictly on the hyperbolic manifold $\mathbb{H}^d$. One could seek the second order approximation to the defining function, which make the robust criterion a quadratic form on the uncertainty parameter $z$.*

**Remark 2.** *Equation (33) arises from a $l_\infty$ uncertainty set. We could of course generalize this analysis to other types of uncertainty sets such as $l_2$ uncertainty and ellipsoidal uncertainty, each with different number of auxiliary variables introduced in their linearization.*

