# OpenReview forum: "Convex Relaxation for Solving Large-Margin Classifiers in Hyperbolic Space"
_TMLR — Accepted by TMLR_

### Review · Reviewer_ZMJ4 · 2024-12-02

**Summary Of Contributions:**

The paper considers minimization problems over a hyperbolic space (instead of the usual euclidean space). They specifically consider the soft-margin SVM as in Cho et al. (2019):

- They apply a Taylor expansion to obtain a (non-convex) quadratically-constrained quadratic programming (QCQP). They consider an SDP relaxation (Eq. 8) and a moment-sum-of-squares relaxation (Eq. 13) for which they develop a sparse variant (Eq. 14).
- They empirically test against projected gradient descent (PGD) on 2-dimensional cases (synthetic Gaussian and tree embeddings, 2d feature embeddings of datasets including CIFAR10/Fashion-MNIST)
- They observe that Sparse-Moment performs well on one-vs-rest classification split (but is slow due to poor scaling in #datapoints), whereas all methods perform much more similarly (and better) on one-vs-one classification split.

**Audience:**

Yes

**Claims And Evidence:**

No

**Requested Changes:**

_Overclaim of empirical contributions_. I suggest either providing a much more extensive comparison, or downtoning the claim in comparison with PGD (specifically the statement "We observe that they achieve better classification accuracy and F1 score than the existing PGD approach on both simulated and real dataset" in the discussion):

- All methods performs much more similarly (and better) on the one-vs-one. I suggest focusing on this case and leaving one-vs-rest as an additional experiment, since it is worse for all methods (performancewise for PGD and computationally for SDP and Sparse-Moment).
- PGD seem to perform better for the largest regularization factor C=10 considered (Table 7). Why not consider a larger factor? It also seem unusual to keep the stepsize fixed regardless of the regularization constant.
- It is stated that "Nevertheless, both relaxed models significantly outperform projected gradient descent (PGD) by a wide margin". Considering that Sparse-Moment is orders of magnitude slower than PGD (Table 11), it seems reasonable to at least tune the stepsize of PGD.
- All methods perform much better on one-vs-one split. Why not consider one-vs-one for the synthetic examples?

The reformulations requires more detail in the paper:

- What are the specific relaxations made to obtain the Moment reformulation (eq. 13) and the SDP (eq. 8).
- Is Eq. 14 an equivalent reformulation of Eq. 13? This seem to be suggest by the statement "we observe a star-shaped sparsity".
- Can Eq. 7 formally be related to the soft-margin in Eq. 6?

Vague statements:

- It is unclear what "Moment" in the experiments specifically refers to. Does "sparse moment-sum-of-squares relaxations (Moment)" in the experiments refers to Eq. 14?
- "However, moment-sum-of-squares relaxation does not scale with the data size due to the combinatorial factors". From the experiments Sparse-Moment do no scale well in datasize either (Table 11). If would be helpful if the authors could compare the computational complexity of Sparse-Moment and moment-sum-of-squares.
- "their hyperbolic embeddings obtained through standard hyperbolic embedding procedure". It would be helpful if the authors elaborate on the procedure to make the paper self-contained.
- "Its exponential volume growth with respect to radius motivates representation learning of hierarchical data using the hyperbolic space". Could the authors elaborate?

Typos:

- datasetsâĂŤfootball, âĂŤto

**Strengths And Weaknesses:**

**Strengths**:

- The paper considers a reformulation of hyperbolic soft-margin SVM that can be inputted into standard solvers.
- The paper is transparently written, discussing various tradeoffs and limitations of the methods.
- Related work is properly cited.

**Weaknesses**:

- Theoretically not much stronger guarantees than PGD: i) SDP only matches originally formulation when low rank. ii) Sparse-Moment only has guarantees when the flat-extension property is satisfied (which is not the case for the given problem according to the authors)
- Empirically the advantage over PGD is not clear: i) on one-vs-one the empirical performance is similar (and PGD could be further tuned) ii) The proposed methods are computationally much slower especially for large #datapoints.
- Empirical evaluation is limited to 2 dimensional examples

---

> ### Author Response · Authors · 2025-01-27
>
> Thanks for the detailed reviews, and we apologize for some of the confusion. Wee would like to address with the followings.
>
> - About overclaiming: We will downtone our conclusion to make it more explicit and precisely aligned with our current empirical evidence. Thank you for suggesting additional experiments (e.g., tuning PGD stepsize, larger regularization factors, one-vs-one synthetic examples). However, due to time and computational constraints, we may defer these to future work.
>
> - About more details in the reformulation:
>     - What are the relaxation made to obtain Eq. 8 and Eq. 14: for Eq.8, note that Eq. 7 is equivalent to Eq. 8 if we require $\boldsymbol{W} = \boldsymbol{w}\boldsymbol{w}^T$ (which implies $\boldsymbol{W}$ is rank -1,) instead of $\boldsymbol{W} - \boldsymbol{w}\boldsymbol{w}^T \succeq 0$; the relaxation done for Eq. 13 is detailed in Appendix C.1, where we truncate the degree of the polynomial to within $2\kappa$ when constructing the Q-module, which is later used to formulate the Eq. 13 through dualization.
>     - Is Eq. 14 equivalent to Eq. 13? Yes! You may think of this as dropping redundant decision variables by exploiting implicit equality constraints in Eq. 13. This is done by observing that the polynomial coefficients $\text{vec}(p)$ contains a lot of 0s in Eq. 13 (e.g. the coefficients for terms $\xi_i \xi_j, \forall i \ne j$ are all 0).
>     - Can Eq. 7. be formally linked to Eq. 6? This is a good point. It is unclear how the Taylor expansion impacts the problem's optimality. However, we would like to point out that both arcsinh and its linearization can serve as valid choices of penalty functions for misclassification (and thus making them more or less like hyperparameters), though arguably the arcsinh formulation is more suited to the hyperbolic space as it scales with hyperbolic distances. Lastly, we would like to note that without making it a polynomial, SDP and Moment relaxation methods would not be accessible.
>
> - About vague statements:
>     - Does (Moment) refer to Eq. 14? Yes. In all experiments, we use the sparse-version Moment SOS to obtain the solutions.
>     - It seem the sparse version also does not scale well either? We acknowledge that indeed even the sparse version suffers from scalability issues. However we will add a detailed discussion to the paper of the computational complexity improvements from the vanilla version to the sparse version.
>     - How are hyperbolic embeddings obtained? Thanks for pointing this out. We will also add clear explanations on these embeddings to make the paper self-contained.
>     - What is the relation between exponential growth and hierarchical representation learning? If we think of a hierarchical dataset as a tree, and observe that typically the number of neighbors of a tree node grows exponentially as the number of hops increases, then the fact that volume grows exponentially with respect to radius in hyperbolic space makes it a natural choice for embedding tree-like dataset, compared to Euclidean spaces of any dimension that only grows polynomially. We will elaborate on this in the later version.
>
> - We apologize for these formatting issues. We will fix them and ensure consistency throughout the paper.

---

> > ### Comment · Reviewer_ZMJ4 · 2025-02-07
> >
> > I thank the authors for their rebuttal.
> >
> > Considering that no more experiments will be considered, I strongly recommend instead explicitly discussing the limitations of the experimental comparison (baselines are not tuned properly, the one-vs-one might lead to a very different conclusion etc).
> > I would recommendation down-toning the empirical claims throughout the paper not just the conclusion.

---

### Review · Reviewer_dQpE · 2024-12-09

**Summary Of Contributions:**

The paper targets the problem of efficient and accurate machine learning in hyperbolic space. In particular, given the embeddings of a dataset in hyperbolic space, the paper targets the problem of machine learning within the hyperbolic space itself, which is in contrast to carrying out machine learning in Euclidean space. The specific ML problem of interest is hyperbolic SVM. Compared to prior works where hyperbolic SVM is addressed using the projected gradient descent (PGD) method, the paper proposes a different optimization approach that seeks to overcome the limitations of PGD in terms of general sensitivity to hyperparameters and initializations. Multiple experiments demonstrate improved performance of the proposed framework over PGD approach for quite a few datasets.

**Audience:**

Yes

**Broader Impact Concerns:**

The reviewer doesn't have any concern that would require a Broader Impact Statement.

**Claims And Evidence:**

No

**Requested Changes:**

•	Please provide in-depth discussion on the proposed hyperbolic SVM framework in [1]. Specify what the limitations are of [1] and how the current manuscript overcomes those limitations.

•	Provide thorough experimental comparisons with [1].

•	Generalize the paper to when curvature is not -1.

•	Vector notations are not consistent. For example, in Section 3.1, in Hyperbolic Space definition, why italic and non-italic notations are used to represent the same variable? Then, in Tangent Space definition, vectors are represented by non-bold font. Please make all notations consistent.

•	In Table 1, in some cases, PGD performs very close to SDP and Moment. In fact, for Gaussian 2, PGD is better than SDP. Furthermore, there are cases where SDP is same as Moment. Please provide intuition behind these results. How can a practitioner ascertain beforehand which method should be used?

**Strengths And Weaknesses:**

Pros:

•	The problem under consideration is interesting and relevant to machine learning over hierarchical datasets.

•	The proposed framework is explained in detail.

•	The paper includes multiple experiments.


Cons:

•	In the prior work of Chien et al. (2021), the authors do not follow a PGD approach to the best of my understanding. In fact, their main contribution is to provide a reformulation of hyperbolic SVM (albeit in the Poincare ball model of hyperbolic space) into an equivalent convex optimization problem. You may also check [1], a follow up to Chien et al. (2021), for more detailed information. Furthermore, [1] also shows that their proposed approach is highly scalable, and provides much better performance than prior works, some of which may not even converge in many settings considered in [1].

•	There is no performance comparison with Euclidean baselines. Particularly, what is the performance of (a) Euclidean SVM over Euclidean embeddings, (b) Euclidean SVM over hyperbolic embeddings, (c) hyperbolic SVM over Euclidean embeddings. This is important to ascertain what improvements one is getting when applying hyperbolic embedding or/and hyperbolic machine learning.

•	There is no discussion on how one can jointly optimize hyperbolic embedding and train a hyperbolic SVM model on the hyperbolic embeddings. I acknowledge that this has been largely ignored in the prior works as well.

•	The curvature of hyperbolic space is negative, but not necessarily -1. The framework seems to only apply when curvature is -1.

[1] Pan, Chao, Eli Chien, Puoya Tabaghi, Jianhao Peng, and Olgica Milenkovic. "Provably accurate and scalable linear classifiers in hyperbolic spaces." Knowledge and Information Systems 65, no. 4 (2023): 1817-1850.

---

> ### Author Response · Authors · 2025-01-27
>
> Thank you for your valuable questions. We would like to make the following comments:
>
> - How does our method compare to Chien (2021)? You are correct that they did not use any gradient descent based algorithm. From our understanding their hyperbolic SVM differs significantly with ours and from Cho (2019). The learning procedure in Chien (2021) can be described as follows:
>     - find the convex hull of each class using a deterministic algorithm that scales as $O(N \log N)$, where $N$ is the number of data points
>     - given respective convex hulls, find a reference point as the midpoint of the minimum distance pairs of points from each convex hull
>     - given the reference point, the problem now becomes convex, since we may project all points to the tangent space of the reference point and everything becomes Euclidean again.
>
>     It is a great algorithm that works well in practice.  However, Chien et al. (2021) focus on reformulating the problem into a Euclidean one rather than directly solving the "original" formulation proposed in Cho et al. (2019). While their algorithm provides a globally optimal solution given a reference point, it is unclear whether **the algorithm *AND* the reference point** are jointly globally optimal with respect to Cho et al. (2019)'s formulation.
>
>     In contrast, our work aims to to solve Cho et al. (2019)'s formulation directly (in Lorentz manifold) using polynomial optimization techniques. Our approach seeks guarantees of global optimality or empirically small optimality gaps, as demonstrated in the paper. However, we acknowledge that this comes at the cost of computational efficiency compared to Chien et al. (2021). Since our methods and Chien et al. (2021) address different problem formulations, we did not make direct comparisons in our paper.
>
> - Comparisons to Euclidean baseline? This is a very good point, but we think this is beyond the scope of this paper, as this also involves the question of obtaining *equivalently expressive representation* in the Euclidean space as in the Hyperbolic space, before training classification in respective manifolds.  Our paper focuses on a narrower question: can Cho et al. (2019)'s formulation be solved directly in the Lorentz manifold, given hyperbolic features?
>
> - This ties to one of your other questions as well that if we could jointly train embeddings and the decision boundaries. This is indeed an interesting but under-explored topic in the literature. We would like to propose using neural network with multi-headed supervision towards this purpose: train a neural network whose representation space is hyperbolic and some linear/mlp heads attached to the representation space for logits and other heuristics such as maximize distance across classes and minimize distances in the same classes, encouraging neural collapses.  However, we consider this topic is outside the scope of our paper and suggest it as a future research direction.
>
> - Can we generalize to curvatures other than -1? Certainly! We would add discussion of extending to beyond just -1. Following Table 1 in paper "Mixed-Curvature Variational Autoencoders" (Skopek et al.) and retracing the proof in the supplemental material in [1], we see that the decision function, the objective, and many constraints stay the same except the following slack variable constraint in Equation 16:
>
> $$
>         \xi_i \geqslant \frac{1}{c}\text{arcsinh}(1) - \frac{1}{c} \text{arcsinh}(-(b^{(i)})^T \boldsymbol{w}), \forall i \in [n],
> $$
> and with some rearrangements, we have  $
>         -(b^{(i)})^T \boldsymbol{w} \geqslant \text{sinh}(\text{arcsinh}(1) - c\xi_i) = \frac{1 - \sqrt{2}}{2}e^{c\xi_i} + \frac{1 + \sqrt{2}}{2}e^{-c\xi_i} =: g_c(\xi_i), $ whose Taylor expansion around $\xi_i = 0$ gives
> $
>         g_c(\xi_i) \approx 1 - \sqrt{2}c\xi_i.
> $
> This effectively introduces a scaling factor $c$ in the formulation. We will include this generalization in the later version.
>
> - About inconsistent notations: we sincerely apologize for the inconsistencies and confusions. We will make everything consistent in the later version of our paper.
>
> - How do practisioner know which method to use in advance? This is an excellent question, and we acknowledge that our current intuition is preliminary. Based on our observations:
>
>     - For PGD vs semidefinite programming (SDP and Moment SOS), the extent of class mixing in the data seems to be a key factor. Arguably Gaussian 2 has less of a mixing than Gaussian 1 and 3, allowing PGD to perform better. More mixing likely leads PGD to poor local minima. However, this hypothesis requires further quantification.
>
>     - For SDP and Moment SOS, by comparing Table 2 and Table 3, SOS tends to perform better in the one-vs-rest scheme, while SDP works well in the one-vs-one scheme. This might suggest that SOS can handle more complex datasets effectively.
> We plan to investigate these intuitions further in future work and provide more concrete recommendations for practitioners.

---

### Review · Reviewer_dPiK · 2025-01-02

**Summary Of Contributions:**

This paper studies the optimization problem of training a support vector machine on a Lorentz manifold. The authors show that the problem can be formulated as a non-convex quadratically-constrained quadratic programming problem, and apply two known convex relaxations for solving it, namely semi-definite programming (SDP) and moment-sum-of-squares (Moment). Due to the prohibitive complexity of the Moment relaxation, the authors employ the sparsity of the problem and consider a sparse version of the Moment relaxation with smaller complexity. The authors demonstrate via experiments on synthetic and real datasets that these reductions typically perform better than projected gradient descent which was used in prior works.

**Audience:**

Yes

**Claims And Evidence:**

Yes

**Requested Changes:**

* I could not follow why we would choose larger $\kappa$ for the Moment relaxation applied to this problem. Isn't the original polynomial $p(q)$ a second-degree polynomial, for which it would suffice to have $\kappa = 1$?

* Given that there is some space left in the main text, the authors could add more details in the main text as opposed to the appendix in how the constraints and solutions of the Moment relaxation relate to the constraints and solutions of the original problem, which would help readers who do not have a background in moment optimization.

* The notation in Equations (10) and (11) is a bit misleading. If I understand correctly, $\langle f_z, A(q)\rangle$ for some matrix of polynomials $A(q)$ means applying $f_z$ element-wise to $A(q)$, whereas the notation suggests some sort of inner product between $f_z$ and $A(q)$. Perhaps the authors can use a different notation here, while keeping $\langle f_z, p\rangle$ between polynomials.

* There are some unreadable characters in the Dataset paragraph of page 7.

* Parentheses are missing for citations that are not part of the sentence such as Mishne et al. (2023) and Weber et al. (2020) in the second paragraph of page 1.

**Strengths And Weaknesses:**

I am not an expert in this area and have not followed the relevant literature, so my evaluation is somewhat limited.

### Strengths:
* The paper provides extensive numerical experiments.
* The paper is generally well-written and is mostly easy to understand.
* In principle and based on the experiments, for low-dimensional problems, the introduced approach will better handle the non-convexity of the problem and outperform gradient-based approaches.

### Weaknesses:
* As the authors also mention in the manuscript, a major weakness is that unlike gradient-based algorithms, the relaxations here scale poorly with problem dimension.
* The relaxations introduced here still lack convergence guarantees similar to gradient-based algorithms.

---

> ### Author Response · Authors · 2025-01-27
>
> Thank you for your insightful questions. We would like to address with the following.
>
> - Why not $\kappa = 1$? The goal of moment sum-of-square relaxtion in a high level is to decompose a polynomial into a sum of square of polynomials. Although the original problem is quadratic, it may not be written as a sum of square of polynomials up to degree 2. In such cases, higher-order terms are introduced, with the hope that they cancel each other at larger $\kappa$. As detailed in Appendix C.1, Lasserre (2001) showed that when $\kappa = \infty$,  there is an exact equivalence between the original polynomial problem and its SOS relaxation. We appreciate your insight and encourage readers to refer to this appendix for more details.
>
> - Suggestion of making more moment relaxation materials to the main text: Thank you for pointing this out. We will consider moving additional details on the constraints and solutions of the moment relaxation from the appendix to the main text in a future revision upon acceptance.
>
> - Clarification on notation: You are absolutely correct that the angle bracket here denote an inner product when applied to vectors and elementwise application when applied to polynomials. We followed the notation from Chapter 5.1 of (Yang, https://hankyang.seas.harvard.edu/Semidefinite/Moment.html). To avoid confusion, we will revise the notation in future versions while maintaining clarity and consistency.
>
> - Missing characters and citations: we apologize for these formatting issues and will make sure they are corrected in the later version of this paper.

---

### Decision · Action_Editor_MHuV · 2025-03-08

**Recommendation:** Accept with minor revision

**Comment:**

The paper proposes algorithms for solving the SVM problem in hyperbolic space by considering convex relaxations based on semi-definite programming and sum-of-squares. The authors then compare the proposed approach to previously studied approaches, mainly projected gradient descent.

The reviewers appreciated the problem, the proposed algorithms, and the conducted empirical analysis. Therefore, the decision is leaning towards acceptance. Nevertheless, reviewers expressed concerns on the empirical study, particularly regarding the limited comparisons with other baselines and restriction to low-dimensional datasets.

Given that the authors did not update the manuscript during the rebuttal, I ask that the authors address the requested changes in a revised manuscript. The revision should also include both a clear discussion and a rigorous experimental study of when the proposed methods perform better or worse than an appropriately tuned PGD baseline.

**Audience:**

Yes

**Claims And Evidence:**

The theoretical claims are accurate and clear. The claims in the empirical study could be made more clear in the revision.